# Proximal Causal Inference with Text Data

**Jacob M. Chen**
Department of Computer Science
Johns Hopkins University
jchen459@jhu.edu

**Rohit Bhattacharya**
Department of Computer Science
Williams College
rb17@williams.edu

**Katherine A. Keith**
Department of Computer Science
Williams College
kak5@williams.edu

## Abstract

Recent text-based causal methods attempt to mitigate confounding bias by estimating proxies of confounding variables that are partially or imperfectly measured from unstructured text data. These approaches, however, assume analysts have supervised labels of the confounders given text for a subset of instances, a constraint that is sometimes infeasible due to data privacy or annotation costs. In this work, we address settings in which an important confounding variable is completely unobserved. We propose a new causal inference method that uses two instances of pre-treatment text data, infers two proxies using two zero-shot models on the separate instances, and applies these proxies in the proximal g-formula. We prove, under certain assumptions about the instances of text and accuracy of the zero-shot predictions, that our method of inferring text-based proxies satisfies identification conditions of the proximal g-formula while other seemingly reasonable proposals do not. To address untestable assumptions associated with our method and the proximal g-formula, we further propose an odds ratio falsification heuristic that flags when to proceed with downstream effect estimation using the inferred proxies. We evaluate our method in synthetic and semi-synthetic settings—the latter with real-world clinical notes from MIMIC-III and open large language models for zero-shot prediction—and find that our method produces estimates with low bias. We believe that this text-based design of proxies allows for the use of proximal causal inference in a wider range of scenarios, particularly those for which obtaining suitable proxies from structured data is difficult.

## 1 Introduction

Data-driven decision making relies on estimating the effect of interventions, i.e. *causal effect estimation*. For example, a doctor must decide which medicine she will give her patient, ideally the one with the greatest effect on positive outcomes. Many causal effects are estimated via randomized controlled trials—considered the gold standard in causal inference; however, if an experiment is unfeasible or unethical, one must use observational data. In observational settings, a primary obstacle to unbiased causal effect estimation is confounding variables, variables that affect both the treatment (e.g., which medicine) and the outcome.

Recently, some studies have attempted to mitigate confounding by incorporating (pre-treatment) unstructured text data as proxies for confounding variables or by specifying confounding variables as linguistic properties, e.g., topic (Veitch et al., 2020; Roberts et al., 2020), tone (Sridhar and Getoor, 2019), or use of specific word types (Olteanu et al., 2017). A wide range of fields have used text in

38th Conference on Neural Information Processing Systems (NeurIPS 2024).

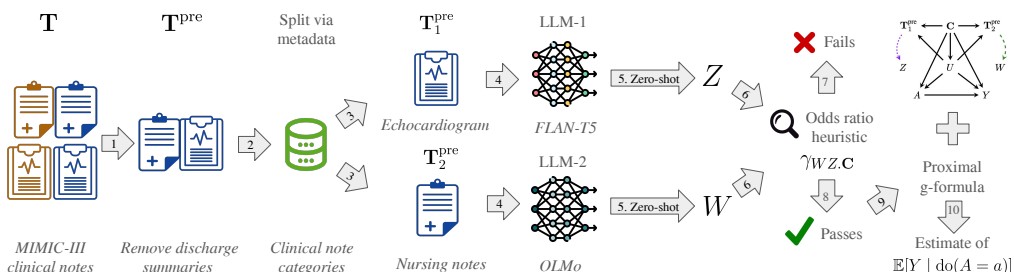

Figure 1: **Pipeline for proximal causal inference with text data**. The top row of captions describe the general pipeline that uses text data from any setting, and the bottom italicized row describes an illustrative example based on our semi-synthetic experiments in Sec. 5. (1) We filter to only pre-treatment text; (2 and 3) for each individual in the analysis, we select two distinct instances of text (e.g., echocardiogram and nursing notes) via metadata with the goal of satisfying $\mathbf{T}_1^{\text{pre}} \not\perp\!\!\!\perp \mathbf{T}_2^{\text{pre}} \mid U, \mathbf{C}$; (4 and 5) we use $\mathbf{T}_1^{\text{pre}}$ and $\mathbf{T}_2^{\text{pre}}$ as inputs into LLM-1 and LLM-2, respectively, to infer zero-shot proxies $Z$ and $W$. (7) If the proxies fail our odds ratio heuristic, analysis stops. (8, 9, and 10) Else, we use the proximal g-formula implied by the casual DAG to estimate the causal effect.

causal estimates, including medicine (Zeng et al., 2022), the behavioral social sciences (Kiciman et al., 2018), and science-of-science (Zhang et al., 2023). See Keith et al. (2020); Feder et al. (2022); Egami et al. (2022) for general overviews of text-based causal estimation.

If all confounders are directly observed, then causal estimation is relatively[1] straightforward with *backdoor adjustment* (Pearl, 2009). However, known confounders are often unobserved. In such scenarios, researchers typically use supervised classifiers to predict the confounding variables from text data, but these text classifiers rarely achieve perfect accuracy and *measurement error* must be accounted for. To address this, another line of work has developed post-hoc corrections of causal estimates in the presence of noisy classifiers (Wood-Doughty et al., 2018; Fong and Tyler, 2021; Egami et al., 2023; Mozer et al., 2023). These approaches, however, require ground-truth labels of the confounding variables for a subset of instances, a constraint that is not always feasible due to privacy restrictions, high annotation costs, or lack of expert labor for labeling.

Our work fills this gap. We address the causal estimation setting for which a practitioner has specified a confounding variable that is truly unmeasured (we have no observations of the variable), but unstructured text data could be used to infer proxies. For this setting, our method combines *proximal causal inference* with zero-shot classifiers.

Proximal causal inference (Miao et al., 2018; Tchetgen Tchetgen et al., 2020; Liu et al., 2024) can identify the true causal effect given *two* proxies for the unmeasured confounder that satisfy certain causal identification conditions. A major criticism of this method is that it can be difficult to find two suitable proxies among the structured variables; however, we conjecture that unstructured text data (if available) could be a rich source of potential proxies.

In our proposed method, summarized in Figure 1, we estimate two proxies from text data via zero-shot classifiers, i.e. classifiers that perform an unseen task with no supervised examples. In subsequent sections, we expand upon this method and its assumptions and empirically validate it on synthetic and semi-synthetic data with real-world clinical notes. Since large pre-trained language models (LLMs) have promising performance on zero-shot classification benchmarks (Yin et al., 2019; Brown et al., 2020; Wei et al., 2021; Sanh et al., 2021, *inter alia*), we use LLMs to infer both of the proxies in our experimental pipeline. Our combination of proximal causal inference and zero-shot classifiers is not only novel, but also expands the set of text-specific causal designs available to practitioners.[2]

In summary, our **contributions** are

---

[1] Setting aside challenges of high-dimensional covariate selection for causal estimation (Tamarchenko, 2023).

[2] Supporting code is available at `https://github.com/jacobmchen/proximal_w_text`.

- We propose a new causal inference method that uses distinct instances of pre-treatment text data, infers two proxies from two different zero-shot models on the instances, and applies the proxies in the proximal g-formula (Tchetgen Tchetgen et al., 2020).
- We provide theoretical proofs that our method satisfies the identification conditions of *proximal causal inference* and prove that other seemingly reasonable alternative methods do not.
- We propose a falsification heuristic that uses the odds ratio of the proxies conditional on observed covariates as an approximation of the (untestable) proximal causal inference conditions.
- In synthetic and semi-synthetic experiments using MIMIC-III's real-world clinical notes (Johnson et al., 2016), our odds ratio heuristic correctly flags when identification conditions are violated. When the heuristic passes, causal estimates from our method have low bias and confidence intervals that cover the true parameter; when the heuristic fails, causal estimates are often biased.

## 2  Problem Setup And Motivation

To motivate our approach, imagine we are an applied practitioner tasked with determining the effectiveness of thrombolytic (clot busting) medications relative to blood thinning medications to treat clots arising from an ischemic stroke. Such medications are usually administered within three hours of the stroke to improve chances of patient recovery (Zaheer et al., 2011). Given the urgency and the short treatment window, running a randomized experiment to compare these drugs is difficult. Left with only observational data, we examine electronic health records (EHRs) from a database like MIMIC-III (Johnson et al., 2016).

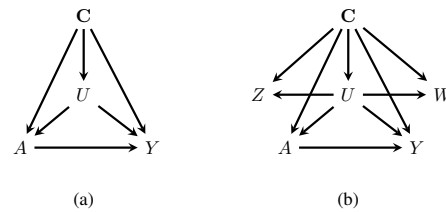

Figure 2: Causal DAGs (a) depicting unmeasured confounding and (b) compatible with the canonical assumptions used for *proximal causal inference* (Tchetgen Tchetgen et al., 2020).

We formalize our causal estimand as follows: let $A$ denote a binary treatment variable corresponding to clot busting ($A = 1$) or blood thinning ($A = 0$) medication, and let $Y$ denote measurements of the D-dimer protein in the patient's blood which directly measures how much of the clotting has dissolved. In do-calculus notation (Pearl, 2009), the target causal estimand is the average causal effect, ACE $:= \mathbb{E}[Y \mid \mathrm{do}(A = 1)] - \mathbb{E}[Y \mid \mathrm{do}(A = 0)]$.

Examining the EHRs, we find potential confounders (in structured tabular form), including biological factors, such as age, sex, and blood pressure, as well as socio-economic factors, such as income. We denote the observed confounders as the set $\mathbf{C}$. However, we are worried about biased causal effects because atrial fibrillation (irregular heart rhythms) is an important confounder corresponding to a pre-existing heart condition that is not recorded in the structured data. We denote this unmeasured confounder as $U$, and assume for the rest of this paper that $\mathbf{C}$ and $U$ form a sufficient backdoor adjustment set with respect to $A$ and $Y$ (Pearl, 1995). Figure 2(a) depicts this problem setup in the form of a causal directed acyclic graph (causal DAG) (Spirtes et al., 2000; Pearl, 2009). With the presence of $U$, it is well known that adjusting for just the observed confounders via the backdoor formula $\sum_{\mathbf{c}} (\mathbb{E}[Y | A = 1, \mathbf{c}] - \mathbb{E}[Y | A = 0, \mathbf{c}]) \times p(\mathbf{c})$ will give a biased estimate of the ACE (Pearl, 1995). In response to this issue, we consider work that uses proxy variables of the unmeasured confounder. However, we are subject to the following **restriction**:

**(R1)** We do not have access to the value of $U$ for any individuals in the dataset.

This kind of restriction is common in healthcare or social science settings when data privacy issues, high costs, or lack of expert labor make analysts unable to hand-label unstructured text data. We elaborate in Appendix A.

In such cases, we turn to using proxies for $U$. Pearl (2010) proposed a method for obtaining unbiased effect estimates with a single proxy $W$ under the assumption that $p(W|U)$ is known or estimable, which are impossible for us under (R1). However, a more recent line of work, building from Griliches (1977) and Kuroki and Pearl (2014), called *proximal causal inference* (Miao et al., 2018; Tchetgen Tchetgen et al., 2020) is able to identify the true causal effect as long as the analyst proposes two proxies $W$ and $Z$ that satisfy the following independence conditions:

**(P1)** Conditional independence of proxies: $W \perp\!\!\!\perp Z \mid U, \mathbf{C}$

**(P2)** One of the proxies, say $W$, does not depend on values of the treatment: $W \perp\!\!\!\perp A \mid U, \mathbf{C}$

**(P3)** The other proxy, $Z$, does not depend on values of the outcome: $Z \perp\!\!\!\perp Y \mid A, U, \mathbf{C}$

A canonical example of proxies that satisfy these conditions is shown in Figure 2(b)[3]. In addition to these independence relations that impose the absence of certain edges in the causal DAG, e.g., no edge can be present between $Z$ and $W$ to satisfy (P1), there is an additional completeness condition that imposes the existence of $U \to Z$ and $U \to W$. This condition is akin to the relevance condition in the instrumental variables literature Angrist et al. (1996) and ensures that the proxies $W$ and $Z$ exhibit sufficient variability relative to the variability of $U$.

**(P4)** Completeness: for any square integrable function $v(\cdot)$ and for all values $w, a, \mathbf{c}$, we have
$$\mathbb{E}[v(U) \mid w, a, \mathbf{c}] = 0 \iff v(U) = 0, \text{ and } \mathbb{E}[v(Z) \mid w, a, \mathbf{c}] = 0 \iff v(Z) = 0.$$

Intuitively, these completeness conditions do not hold unless $Z$ and $W$ truly hold some predictive value for the unmeasured confounder $U$. See Miao et al. (2018) for more details on completeness. Under (P1-P4), each piece of the ACE, $\mathbb{E}[Y \mid \mathrm{do}(A = a)]$, is identified via the *proximal g-formula*,

$$\mathbb{E}[Y \mid \mathrm{do}(a)] = \sum_{w, \mathbf{c}} h(a, w, \mathbf{c}) \times p(w, \mathbf{c}), \tag{1}$$

where $h(a, w, \mathbf{c})$ is the "outcome confounding bridge function" that is a solution to the equation $\mathbb{E}[Y \mid a, z, \mathbf{c}] = \sum_w h(a, w, \mathbf{c}) \times p(w \mid a, z, \mathbf{c})$ (Miao et al., 2018). Although the existence of a solution is guaranteed under (P1-P4), solving it can still be difficult. However, there exist simple two-stage regression estimators for the proximal g-formula (Tchetgen Tchetgen et al., 2020; Mastouri et al., 2021) that we make use of in Section 5 once we have identified a valid pair of proxies. This brings us to the primary criticism of proximal causal inference.

**Primary criticism**   When using proximal causal inference in practice, how do we find two proxies $W$ and $Z$ among the structured variables such that they happen to satisfy all of (P1-P4)? We often cannot, at least not without a high degree of domain knowledge.[4] Furthermore, empirically testing for any of (P1-P3) in general is, by definition of the problem, not possible because doing so requires complete access to the unobserved confounder $U$.

**Our approach**   Instead, in this work, we propose relying on raw unstructured text data (e.g., clinical notes) in an attempt to infer proxies that satisfy (P1-P4) ***by design***.

Returning to our motivating problem, our approach applies classifiers zero-shot—since we have no training data with $U$ given (R1)—to two distinct instances of pre-treatment clinical notes of each patient and obtains two predictions, $W$ and $Z$, for atrial fibrillation (our $U$). In order to make it more feasible to estimate the ACE from data, we make the following two relatively weak assumptions:

**(S1)** The unmeasured confounder $U$ between $A$ and $Y$ can be specified as a binary variable.

**(S2)** The text only causes $W$ and $Z$ (and no other variables).

Assuming (S1) simplifies estimation since text classification typically performs better empirically than text regression (Wang et al., 2022). Assumption (S2) asserts that the text data considered serves as a record of events rather than actionable data.[5]

---

[3]One could also add the edges $W \to Y$ and $Z \to A$ to Figure 2(b), but these relations will not show up in our text-based setting. Shpitser et al. (2023) also propose a general proximal identification algorithm that is compatible with other causal DAGs, but we focus on the canonical proximal learning assumptions stated in Tchetgen Tchetgen et al. (2020).

[4]For instance, suppose we aim to find proxies (in the structured variables) of atrial fibrillation ($U$) and have access to shortness of breath and heart palpitations. Although these proxies seem reasonable, we show how they may violate the proximal causal inference conditions. Patient complaints about shortness of breath may affect which medication a clinician prescribes ($A$); hence, using shortness of breath as the proxy $W$ violates (P2). Furthermore, a lack of oxygen resulting from shortness of breath may affect the healing of blood clots, thus influencing measurements of the D-dimer protein $Y$. In this case, using shortness of breath as the proxy $Z$ violates (P3). Shortness of breath can also sometimes be a symptom of heart palpitations, violating (P1).

[5]We present an example of how to relax this assumption in Appendix B. We leave to future work more complicated scenarios, e.g., if the reader's perception of text differs from the writer's intent Pryzant et al. (2021).

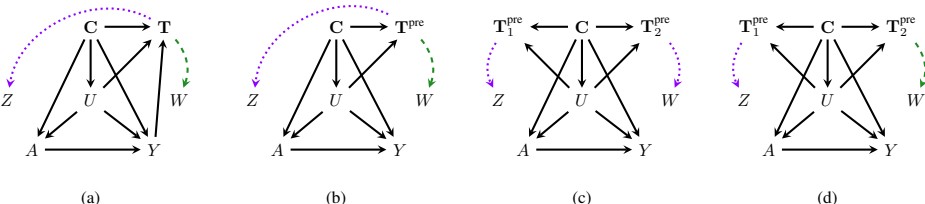

Figure 3: Causal DAGs depicting several different scenarios for inferring text-based proxies. Edges with different colors and patterns, e.g., $\mathbf{T} \dashrightarrow \mathbf{Z}$ and $\mathbf{T} \dashrightarrow \mathbf{W}$, indicate that different zero-shot models were used. Our final recommended method is based on (d).

## 3  Designing Text-Based Proxies

In this section, we describe our method for designing text-based proxies. In doing so, we describe various "gotchas," pitfalls in attempting to use these text-based proxies in causal effect estimation. We describe each gotcha and explore how they lead us to our final recommended design, given by Figure 3(d). Our empirical results in Section 5 demonstrate, as expected, that these pitfall approaches result in biased causal effect estimates.

**Gotcha #1: Using predictions directly in backdoor adjustment.**   Suppose we try to avoid the complications of proximal causal inference by using the predictions from one of our zero-shot models, say $W$, as the confounding variable itself.

**Proposition 1.** *Using a proxy $W$ in the backdoor adjustment formula results in biased estimates of the ACE in general.*

*Proof.* If $W \neq U$ for some subset of instances, there remains an open backdoor path through $U$, and the ACE remains biased as $\sum_{U,\mathbf{C}}(\mathbb{E}[Y \mid A = 1, U, \mathbf{C}] - \mathbb{E}[Y \mid A = 0, U, \mathbf{C}]) \times p(U, \mathbf{C}) \neq \sum_{W,\mathbf{C}}(\mathbb{E}[Y \mid A = 1, W, \mathbf{C}] - \mathbb{E}[Y \mid A = 0, W, \mathbf{C}]) \times p(W, \mathbf{C})$ in general (Pearl, 1995). $\qquad\square$

The most straightforward way to obtain unbiased results under Gotcha #1 is to have 100% accuracy between $W$ and $U$, a scenario that is extremely unlikely in the real world. Further, under (R1)[6], we cannot measure accuracy at inference time since we do not have any observations of $U$. As expected, in our semi-synthetic experiments in Section 5, we find using predictions of $W$ directly in a backdoor adjustment formula results in biased estimates; see Figure 4.

**Gotcha #2: Using post-treatment text.**   While it is well-known that adjusting for post-treatment covariates in the backdoor formula often leads to bias (Pearl, 2009), it is not obvious what might go wrong when using post-treatment text to infer proxies for the proximal g-formula.

**Proposition 2.** *If both $W$ and $Z$ are inferred from zero-shot models on text that contain post-treatment information, then the resulting proxies violate either (P2), (P3), or both.*

*Proof.* Consider Figure 3(a), where the proxies are produced using text that is post-outcome and thus also post-treatment. We show that this violates both (P2) and (P3). Clearly the DAG violates (P3): by a simple d-separation argument we see that $Z \not\perp\!\!\!\perp Y \mid A, U, \mathbf{C}$ due to the open path $Y \to \mathbf{T} \dashrightarrow Z$. Similarly, (P2) is violated from the open path $A \to Y \to \mathbf{T} \dashrightarrow W$. $\qquad\square$

Thus, before performing zero-shot inference, it is important that the text for each individual is filtered in such a way that it contains only the text preceding treatment[7]. In our running clinical example, we can avoid this gotcha by using the time stamps of the clinical notes and information about when the patient was treated and discharged.

---

[6]In the absence of (R1), we direct readers to work that adjusts via measurement error estimates or assumptions that $U$ is "missing at random" Wood-Doughty et al. (2018).

[7]In Appendix B we show how proxies could be inferred using a mix of pre- and post-treatment text while satisfying (P1-P3). However, we advocate for the the pre-treatment rule due to its simplicity.

**Gotcha #3: Predicting both proxies from the same instance of text.** After filtering to only pre-treatment text, $\mathbf{T}^{\mathrm{pre}}$, for each individual, the intuitive next step is to use $\mathbf{T}^{\mathrm{pre}}$ to infer $W$ and $Z$.

**Proposition 3.** *If $W$ and $Z$ are inferred via zero-shot models on the same instance of pre-treatment text, the resulting proxies violate (P1).*

*Proof.* Consider the causal DAG with proxies $W$ and $Z$ in Fig 3(b). By d-separation we have $W \not\!\perp\!\!\!\perp Z \mid U, \mathbf{C}$ due to the path $Z \dashleftarrow \mathbf{T}^{\mathrm{pre}} \dashrightarrow W$. $\square$

In our synthetic experiments in Section 5, we find inferring proxies from the same instance of text results in biased estimates; see Table 1. To avoid Gotcha #3, we select two distinct instances of pre-treatment text data for each unit of analysis, $\mathbf{T}_1^{\mathrm{pre}}$ and $\mathbf{T}_2^{\mathrm{pre}}$, such that $\mathbf{T}_1^{\mathrm{pre}} \perp\!\!\!\perp \mathbf{T}_2^{\mathrm{pre}} \mid U, \mathbf{C}^8$. For example, in the clinical setting we use metadata to select two separate categories of notes for each individual patient, e.g., nursing notes and echocardiogram notes. We hypothesize this satisfies $\mathbf{T}_1^{\mathrm{pre}} \perp\!\!\!\perp \mathbf{T}_2^{\mathrm{pre}} \mid U, \mathbf{C}$ since different providers will likely write reports that differ in content and style. Although the availability of distinct instances is domain-dependent, we hypothesize this type of data exists in other domains as well; for example, one could select distinct social media posts written by the same individual or multiple speeches given by the same politician.

**Gotcha #4: Using a single zero-shot model.** After splitting text into $\mathbf{T}_1^{\mathrm{pre}}$ and $\mathbf{T}_2^{\mathrm{pre}}$, should we apply the same zero-shot model to these two instances of text to infer the proxies $W$ and $Z$, as in Figure 3(c), or should we apply two separate zero-shot models as in Figure 3(d)? We find different answers in theory and practice. Proposition 4 shows that, in theory, both are valid as long as $\mathbf{T}_1^{\mathrm{pre}} \perp\!\!\!\perp \mathbf{T}_2^{\mathrm{pre}} \mid U, \mathbf{C}$. However, we later describe how our semi-synthetic experiments demonstrate the need for two different models in practice in Section 5. We first establish the theoretical validity of both strategies.

**Proposition 4.** *If $W$ and $Z$ are inferred using zero-shot classification on two unique instances of pre-treatment text such that $\mathbf{T}_1^{\mathrm{pre}} \perp\!\!\!\perp \mathbf{T}_2^{\mathrm{pre}} \mid U, \mathbf{C}$, then these proxies satisfy (P1-P3). Additionally, if the proxies are predictive of $U$, i.e., $Z \not\!\perp\!\!\!\perp U \mid \mathbf{C}$ and $W \not\!\perp\!\!\!\perp U \mid \mathbf{C}$, then (P4) holds.*

*Proof.* Suppose we apply zero-shot classification models to two splits of pre-treatment text in a way that results in causal DAGs shown in Figure 3(c) or (d) depending on whether we use one or two models, respectively. Applying d-separation confirms that the conditions (P1-P3) hold in both cases.

Let $|\mathfrak{X}_V|$ denote the number of categories of a variable $V$. Kuroki and Pearl (2014); Tchetgen Tchetgen et al. (2020) state that when $W$ and $Z$ are predictive of $U$ (as stated in the proposition), a sufficient condition for (P4) is $\min(|\mathfrak{X}_Z|, |\mathfrak{X}_W|) \geq |\mathfrak{X}_U|$. Since $U$ is binary under (S1) and since $W$ and $Z$ are discrete variables because they are inferred from classifiers, this condition is satisfied. Hence, (P4) is satisfied. $\square$

**Our Final Design Procedure** Figure 1 and the causal DAG in Figure 3(d) summarize our final design procedure—obtain two distinct instances of pre-treatment text for each individual and apply two distinct zero-shot models to obtain $W$ and $Z$.

In Proposition 4, we formalized how this procedure can be used to design proxies that satisfy the proximal conditions (P1-P4). However, this result relied on two important pre-conditions: (1) the conditional independence of the two instances of text, and (2) $W$ and $Z$ being (at least weakly) predictive of $U$. Yet, both of these conditions are untestable in general; this motivates the next section.

## 4 Falsification: Odds Ratio Heuristic

In practice, a major challenge for our procedure—and indeed all causal methods—are its assumptions. Sometimes, causal models imply testable restrictions on the observed data that can be used in *falsification* or *confirmation tests* of model assumptions, see Wang et al. (2017); Chen et al. (2023); Bhattacharya and Nabi (2022) for tests of some popular models. In our case, the proximal model

---

[8]Note this independence condition does not imply the two pieces of text are completely uncorrelated. Since the text is written based on observations of the same individual, we certainly expect $\mathbf{T}_1^{\mathrm{pre}} \not\!\perp\!\!\!\perp \mathbf{T}_2^{\mathrm{pre}}$; we simply require that the two pieces are correlated only due to $\mathbf{C}$ and $U$.

---

**Algorithm 1** for inferring two text-based proxies

---

1: **Inputs**: Observed confounders $\mathbf{C}$; Text $\mathbf{T}$; Zero-shot models $\mathcal{M}_1, \mathcal{M}_2$; Specified $\gamma_{\text{high}}$
2: Select two distinct instances of pre-treatment text $\mathbf{T}_1^{\text{pre}}$ and $\mathbf{T}_2^{\text{pre}}$ from $\mathbf{T}$
3: Infer $Z \leftarrow \mathcal{M}_1(\mathbf{T}_1^{\text{pre}})$ and $W \leftarrow \mathcal{M}_2(\mathbf{T}_2^{\text{pre}})$
4: Compute the confidence interval (CI) for $\gamma_{WZ.\mathbf{C}}$, $(\gamma_{WZ.\mathbf{C}}^{\text{CI low}}, \gamma_{WZ.\mathbf{C}}^{\text{CI high}})$
5: **if** $1 < \gamma_{WZ.\mathbf{C}}^{\text{CI low}}$ and $\gamma_{WZ.\mathbf{C}}^{\text{CI high}} < \gamma_{\text{high}}$ **then** return $W$ and $Z$
6: **else** return "stop"

---

implies no testable restrictions (Tchetgen Tchetgen et al., 2020), so the best we can do is provide a *falsification heuristic* that allows analysts to detect serious violations of (P1-P4) when using the inferred proxies and stop analysis. Our heuristic is based on the odds ratio function described below.

Given arbitrary reference values $w_0$ and $z_0$, the conditional odds ratio function for $W$ and $Z$ given covariates $\mathbf{X}$ is defined as Chen (2007), $\text{OR}(w, z \mid \mathbf{x}) = \frac{p(w|z,\mathbf{x})}{p(w_0|z,\mathbf{x})} \times \frac{p(w_0|z_0,\mathbf{x})}{p(w|z_0,\mathbf{x})}$. This function is important because $W \perp\!\!\!\perp Z \mid \mathbf{X}$ if and only if $\text{OR}(w, z \mid \mathbf{x}) = 1$ for all values $w, z, \mathbf{x}$. We summarize this odds ratio as a single free parameter, $\gamma_{WZ.\mathbf{X}}$, and, for the simplicity of our pipeline, we estimate it under a parametric model for $p(W|Z, \mathbf{X})$[9].

Now, we describe our proximal conditions in terms of odds ratio parameters. If (P1-P3) are satisfied, then $W \perp\!\!\!\perp Z \mid U, \mathbf{C}$ and $\gamma_{WZ.U\mathbf{C}} = 1$. Further, if the zero-shot models are truly predictive of $U$, then (P4) is satisfied and $W \not\perp\!\!\!\perp Z \mid \mathbf{C}$, which means that $\gamma_{WZ.\mathbf{C}} \neq 1$. Ideally, we would want to estimate both of these odds ratio parameters to confirm (P1-P4) empirically; however, $\gamma_{WZ.U\mathbf{C}}$ cannot be computed from observed data alone due to (R1).

Using a parameter we can estimate from observed data, $\gamma_{WZ.\mathbf{C}}$, we propose an **odds ratio falsification heuristic** in lines 3-6 of Algorithm 1. Now, we explain why, if this heuristic holds, an analyst can be reasonably confident in using their inferred text-based proxies for estimation.

First, we examine a lower bound on $\gamma_{WZ.\mathbf{C}}$. Based on our previous discussion, if $\gamma_{WZ.\mathbf{C}}$ is close to 1, then we should suspect that one or both of our zero-shot models failed to return informative predictions for $U$. Next, let us treat $\gamma_{WZ.\mathbf{C}}$ as an imperfect approximation of $\gamma_{WZ.U\mathbf{C}}$. Let $W, Z, U$ be binary with reference values $w_0 = z_0 = u_0 = 0$. VanderWeele (2008) proposed the following three conditions under which an odds ratio $\gamma_{WZ.\mathbf{C}}$ that fails to adjust for an unmeasured confounder $U$ is an *overestimate* of the true odds ratio $\gamma_{WZ.U\mathbf{C}}$: (i) $\{U\} \cup \mathbf{C}$ satisfies the backdoor criterion with respect to $W$ and $Z$; (ii) $U$ is univariate or consists of independent components conditional on $\mathbf{C}$; (iii) $\mathbb{E}[W|u, z, \mathbf{c}]$ is non-decreasing in $U$ for all $z$ and $\mathbf{c}$ and $\mathbb{E}[Z|u, \mathbf{c}]$ is non-decreasing in $U$ for all $\mathbf{c}$.

Condition (i) is satisfied from Graph 3(d), and condition (ii) is satisfied by assumption (S1). Finally, condition (iii) is satisfied when our zero-shot models are reasonable predictors of the unmeasured confounder $U$ by the following argument. Notice that $\mathbb{E}[W|u, z, \mathbf{c}] = \mathbb{E}[W|u, \mathbf{c}] = p(W = 1|u, \mathbf{c})$, where the first equality follows from d-separation in Graph 3(d) and the second equality follows from the definition of expectation for binary variable $W$. Then we should expect, if the zero-shot models are reasonably accurate, that $p(W = 1|U = 1, \mathbf{c}) > p(W = 1|U = 0, \mathbf{c})$. Therefore, the first part of condition (iii) is satisfied. Similar logic holds for $\mathbb{E}[Z|u, \mathbf{c}]$.

Hence, we have shown that under ideal conditions that satisfy (P1-P4), we should expect $\gamma_{WZ.\mathbf{C}} > \gamma_{WZ.U\mathbf{C}} = 1$, and we should reject proxies $W$ and $Z$ when we observe an odds ratio $\gamma_{WZ.\mathbf{C}} \leq 1$. Next, we examine the upper bound on $\gamma_{WZ.\mathbf{C}}$. Consider the extreme case where $\gamma_{WZ.\mathbf{C}} = \infty$. This corresponds to a situation where $W = Z$, so (P1) is clearly not satisfied. In general, if $\gamma_{WZ.\mathbf{C}}$ is higher than some threshold $\gamma_{\text{high}}$, corresponding to the maximum association that one could reasonably explain by a single open path through $U$, we should suspect that perhaps the proxies $W$ and $Z$ are associated with each other due to additional paths through other unmeasured variables that make it so that the two instances of text are not independent of each other, i.e., $\mathbf{T}_1^{\text{pre}} \not\perp\!\!\!\perp \mathbf{T}_2^{\text{pre}} \mid U, \mathbf{C}$.

Following standard practice in *sensitivity analysis*, e.g., Liu et al. (2013); Leppälä (2023), we leave it to the analyst to specify the upper bound $\gamma_{\text{high}}$ based on domain knowledge. In our experiments in Section 5, we found that, when the proximal conditions are not satisfied, $\gamma_{WZ.\mathbf{C}}$ far exceeds

---

[9]More generally, the odds ratio can be treated as a finite $p$-dimensional parameter vector $\boldsymbol{\gamma}$ and estimated under semi-parametric restrictions on $p(W|Z, \mathbf{X})$ and $p(Z|W, \mathbf{X})$ (Chen, 2007; Tchetgen Tchetgen et al., 2010).

| Estimation Pipeline | $(\gamma_{WZ.\mathbf{C}}^{\text{CI low}}, \gamma_{WZ.\mathbf{C}}^{\text{CI high}})$ | Est. ACE | Bias | Conf. Interval (CI) | CI Cov. |
|---|---|---|---|---|---|
| P1M | $(1.35, 1.42)^{\checkmark}$ | 1.304 | **0.004** | $(1.209, 1.394)$ | **Yes** |
| P1M, same | $(10^{16}, 10^{16})$ | 1.430 | 0.130 | $(1.405, 1.495)$ | No |
| P2M | $(1.82, 1.94)^{\checkmark}$ | 1.343 | **0.043** | $(1.273, 1.425)$ | **Yes** |
| P2M, same | $(7.9, 8.41)$ | 1.407 | 0.107 | $(1.376, 1.479)$ | No |

Table 1: **Fully synthetic results** with the true ACE equal to 1.3. Here, $\checkmark$ distinguishes settings that passed the odds ratio heuristic from those that failed it, with $\gamma_{\text{high}} = 2$. Corresponding to Gotcha #3, "same" indicates we used the same instance of text to infer both $W$ and $Z$.

| $U$ | $\mathbf{T}_1^{\text{pre}}$ Cat. | $\mathbf{T}_2^{\text{pre}}$ Cat. | $\gamma_{WZ.U\mathbf{C}}$ P1M | $\gamma_{WZ.U\mathbf{C}}$ P2M | $\gamma_{WZ.\mathbf{C}}$ CI P1M | $\gamma_{WZ.\mathbf{C}}$ CI P2M |
|---|---|---|---|---|---|---|
| A-Sis | Echo | Radiology | 1.626 | 1.085 | $(2.381, 2.962)$ | $(1.372, 1.995)^{\checkmark}$ |
| Heart | Echo | Nursing | 2.068 | 1.156 | $(2.298, 2.676)$ | $(1.152, 1.376)^{\checkmark}$ |
| A-Sis | Radiology | Nursing | 2.350 | 1.328 | $(4.337, 5.266)$ | $(2.050, 2.663)$ |

Table 2: **Semi-synthetic odds ratio heuristic** ($\gamma_{WZ.\mathbf{C}}$) as well as the oracle $\gamma_{WZ.U\mathbf{C}}$ for different categories of notes (Cat.). We distinguish settings that passed the odds ratio heuristic ($\checkmark$) from those that failed, with $\gamma_{\text{high}} = 2$.

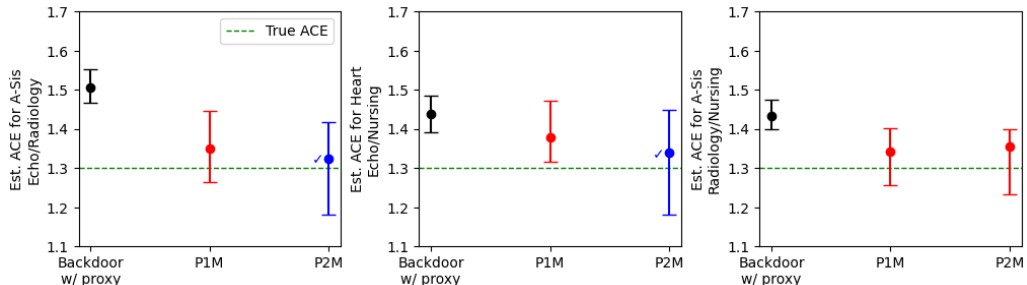

Figure 4: **Semi-synthetic results** for ACE point estimates (dots) and 95% CIs (bars). We distinguish settings that passed the odds ratio heuristic ($\checkmark$) from those that failed, with $\gamma_{\text{high}} = 2$.

any reasonable setting of $\gamma_{\text{high}}$. Hence, our heuristic works quite well in practice even with a generous suggestion for an upper bound. We describe our full design procedure with the diagnostic in Algorithm 1 and calculate 95% confidence intervals for $\gamma_{WZ.\mathbf{C}}$ via the bootstrap percentile method (Wasserman, 2004). We now evaluate its effectiveness for downstream causal inference.

# 5 Empirical Experiments and Results

**RQs** In this section, we explore the following empirical research questions (RQs): How does Algorithm 1 compare to other alternatives in terms of bias and confidence interval coverage of the estimated causal effects? Does our odds ratio heuristic effectively flag when to stop or proceed?

In causal inference, empirical evaluation is difficult because it requires ground-truth labels for counterfactual outcomes of an individual under multiple versions of the treatment, data that is generally impossible to obtain (Holland, 1986). Thus, we turn to synthetic data and semi-synthetic data so we have access to the true ACE and $U$ to evaluate methods. Semi-synthetic experiments—which use real data for part of the DGP and then specify synthetic relationships for the remainder of the DGP—have been used extensively for other empirical evaluation of causal estimation methods; see Shimoni et al. (2018); Dorie et al. (2019); Veitch et al. (2020).[10] We describe the experimental set-ups, the causal estimation procedure used by all experiments, and finally, the results to our RQs.

---

[10]Gentzel et al. (2019) and Keith et al. (2023) describe methods for evaluation that are more realistic than semi-synthetic evaluation by downsampling an RCT dataset to create an observational (confounded) dataset. However, for the proximal setting we investigated in this paper, we could not find a suitable existing RCT.

**Fully Synthetic Experiments**   We create our fully synthetic DGP based on the DAG in Figure 3(d); see Appendix C for full details. To summarize, $A$ and $U$ are binary, and $Y$ and $C$ are continuous. We generate (very simple) synthetic text data with four continuous variables, $X_1, X_2, X_3, X_4$, as functions of $U$ and $C$. For training, we generate two realizations of these variables, which we call $\mathbf{X}_1^{\text{train}}$ and $\mathbf{X}_2^{\text{train}}$, and likewise two realizations for inference time, $\mathbf{X}_1^{\text{inf}}$ and $\mathbf{X}_2^{\text{inf}}$.

At inference time, we explore using one or two zero-shot models, which we refer to as *Proximal 1-Model* (P1M) and *Proximal 2-Model* (P2M), respectively. For one zero-shot model, we train a logistic regression classifier to predict the true $U$ from an aggregated variable $\widetilde{\mathbf{X}}^{\text{train}} = (\mathbf{X}_1^{\text{train}} + \mathbf{X}_2^{\text{train}})/2$ as $P_\theta(U = 1|\widetilde{\mathbf{X}}^{\text{train}})^{11}$. For the other zero-shot model, we use the following heuristic: predict 1 if $X_1 > 1.1$ else 0. P1M uses only the logistic regression model, and P2M uses both the logistic regression and heuristic models. Next, we vary whether Gotcha #3 holds at inference time for both P1M and P2M, i.e. whether the models infer $Z$ and $W$ from only $\mathbf{X}_1^{\text{inf}}$ or both $\mathbf{X}_1^{\text{inf}}$ and $\mathbf{X}_2^{\text{inf}}$.

**Semi-Synthetic Experiments**   For our semi-synthetic experiments, we use MIMIC-III, a de-identified dataset of patients admitted to critical care units at a large tertiary care hospital (Johnson et al., 2016). See Appendices D and G for detailed pre-processing steps, variables, and DGP. To summarize, we use the following real data from MIMIC-III in our DGP: ICD-9 code diagnoses, demographic information, and unstructured text notes. We choose the four diagnoses for oracle $U$ that had the best F1 scores for a supervised bag-of-words logistic regression classifier (see Appendix E for full results): *atrial fibrillation* (Afib), *congestive heart failure* (Heart), *coronary atherosclerosis of the native coronary artery* (A-Sis), and *hypertension* (Hypertension). To avoid Gotcha #2, we explicitly exclude discharge summaries, which are post-treatment. In each experiment, we choose two out of the following four note categories as $\mathbf{T}_1^{\text{pre}}$ and $\mathbf{T}_2^{\text{pre}}$: electrocardiogram (ECG), echocardiogram (Echo), Radiology, and Nursing notes. We hypothesize that notes written about different aspects of clinical care and by different providers will likely satisfy $\mathbf{T}_1^{\text{pre}} \perp\!\!\!\perp \mathbf{T}_2^{\text{pre}} \mid U, \mathbf{C}$ since each individual will write a conditionally independent realization of the patient's status. In Appendix I, we find distinct unigram vocabularies between the note sets that pass our falsification heuristic, lending preliminary evidence to this assumption of conditional independence. Consistent with the DAG in Figure 3(d), we then synthetically generate binary $A$ and continuous $Y$.

For our zero-shot models, we use FLAN-T5 XXL (Flan) Chung et al. (2024) and OLMo-7B-Instruct (OLMo) Groeneveld et al. (2024), both "open" instruction-tuned large language models. In Appendix F, we elaborate on our choice of these models. Following Ziems et al. (2024), we use the prompt template '*Context: {$\mathbf{T}^{pre}$} \nIs it likely the patient has {U}?\nConstraint: Even if you are uncertain, you must pick either "Yes" or "No" without using any other words.*' We assign 1 when the output from Flan or OLMo contains '*Yes*' and 0 otherwise. P1M uses Flan for both proxies $W$ and $Z$ while P2M uses Flan for $W$ and OLMo for $Z$. We compare P1M and P2M to a baseline that uses the inferred proxy $W$ from Flan directly in a backdoor adjustment formula, corresponding to Gotcha #1.

**Estimation of Proximal g-formula**   For all experiments, we estimate the ACE by using the inferred $W$ and $Z$ in a two-stage linear regression estimator for the proximal g-formula provided by Tchetgen Tchetgen et al.. Although the linearity assumption is restrictive, it allows us to focus on evaluating the efficacy of our proposed method for inferring text-based proxies as opposed to complications with non-linear proximal estimation (Mastouri et al., 2021). Briefly, we first fit a linear regression $\mathbb{E}[W|A, Z, \mathbf{C}]$. Next, we infer $\widehat{W}$, continuous probabilistic predictions for $W$, using the fitted model. For the second stage, we fit a linear model for $\mathbb{E}[Y|A, \widehat{W}, \mathbf{C}]$. The coefficient for $A$ in this second linear model is the estimated ACE. We calculate 95% confidence intervals for the ACE via the bootstrap percentile method (Wasserman, 2004). See Appendix F for additional implementation details (e.g., addressing class imbalance and sample splitting).

**Results**   Table 1 has synthetic results, and Tables 2 and Figure 4 have selected results for the semi-synthetic experiments. See Appendices E, H, and J for additional results. In short, our empirical results corroborate preference for Algorithm 1 over the baseline.

First, we discuss the "gotcha" methods we showed to be theoretically incorrect in Section 3. Regarding Gotcha #1, Figure 4 shows that, across all settings of $U$, using the inferred $W$ directly in the backdoor

---

[11]Of course, having access to the true $U$ may not qualify as "zero-shot" in the strict sense of the term, but we complement this idealized scenario with the difficult true zero-shot scenario in our semi-synthetic experiments.

adjustment formula results in estimates with large bias. Regarding Gotcha #3 in the fully synthetic experiments, using the same realization $\mathbf{X}_1^{\mathrm{inf}}$ results in high bias—0.130 and 0.107 for P1M and P2M, respectively, in Table 1—and CIs that do not cover the true ACE. Regarding Gotcha #4, as in Proposition 4, using a single zero-shot model (P1M) results in low bias in the idealized setting of the synthetic DGP (Table 1; first row). However, using real clinical notes and Flan, the second columns of Figure 4 show that P1M produces estimates with higher bias compared to P2M across all three settings. We hypothesize this could be due to Flan over-relying on its pre-training and thus making similar predictions regardless of the clinical note.

For all experiments we set $\gamma_{\mathrm{high}} = 2$. With this setting, we find low bias and good CI coverage for the two settings that pass our heuristic (P2M for $U =$ A-Sis with Echo and Radiology notes and P2M for $U =$ Heart with Echo and Nursing notes) and biased estimates in other cases. Although the confidence intervals sometimes cover the true ACE in settings that fail the heuristic, the interval is skewed and appears will asymptotically converge to a biased value. This gives us confidence that Algorithm 1 appropriately flags when to stop or proceed.

Understanding *why* the odds ratio heuristic fails for a specific application requires oracle data. However, it may be helpful for analysts to reason about scenarios that influence this failure, such as when $\mathbf{C}$ and $U$ are not a sufficient backdoor adjustment set; $\mathbf{T}_1^{\mathrm{pre}} \not\perp\!\!\!\perp \mathbf{T}_2^{\mathrm{pre}} \mid U, \mathbf{C}$; or $Z$ and $W$ are poorly predictive of $U$. Using oracle data, we examined these scenarios for a few settings of our semi-synthetic experiments. For example, Figure 4 (right-most panel) shows that $U=$A-Sis with nursing and radiology notes fails the odds ratio heuristic. We qualitatively examined samples of the clinical notes and found some had unmeasured confounding, e.g., a patient had lung cancer—a variable that was not in our $\mathbf{C}$ or $U$—that influenced descriptions in both $\mathbf{T}_1^{\mathrm{pre}}$ and $\mathbf{T}_2^{\mathrm{pre}}$. Figure 9 (bottom right-most panel) shows that for $U=$hypertension with echocardiogram (echo) and nursing notes also fails the odds ratio heuristic. However, we hypothesize this is due not to text dependence (see Table 4), but rather that the proxies $W$ and $Z$ have accuracies of 0.49 and 0.52 respectively (when evaluated against oracle $U$), which is lower accuracy than predicting the majority class (Table 21). We leave to future work disaggregating the odds ratio heuristic into tests for these specific scenarios.

## 6 Conclusion, Limitations, and Future Work

In this work we proposed a novel causal inference method for estimating causal effects in observational studies when a confounding variable is completely unobserved but unstructured text data is available to infer potential proxies. Our method uses distinct instances of pre-treatment text data, infers two proxies using two zero-shot models on the instances, and applies these proxies in the proximal g-formula. We have shown why one should prefer our method to alternatives, both in theory and in the empirical results of synthetic and semi-synthetic experiments.

If the conditions we present hold, the estimates will be unbiased. Further, as the underlying predictive accuracy of $Z$ and $W$ improve, the confidence interval width will narrow. Thus, we see promise in work improving the zero-shot performance of LLMs. Future work could possibly combine proxies from a greater number of zero shot models, i.e. inspired by work in Yang et al. (2024).

Although we are careful to infer valid proxies by design, it is generally impossible to empirically test whether conditions for proximal causal inference are fulfilled. To address this, we proposed an odds ratio falsification heuristic test, but acceptable values of $\gamma_{\mathrm{high}}$ depend on the domain. If these are set incorrectly, our method could result in false positives (a setting for which the CI does not cover the true parameter and the estimated ACE is biased). In addition, our approach depends on the availability of pre-treatment text data and metadata to split text into independent pieces, which are not available across all applied settings.

Although we use a clinical setting as our running example, our method is applicable to many other domains where it is infeasible to obtain ground-truth $U$ labels due to privacy constraints or annotation costs, e.g., social media or education studies with private messaging data or sensitive student coursework. Other directions for future work include incorporating non-linear proximal estimators (including ones that use alternative assumptions to the completeness condition in (P4) (Kallus et al., 2021)), expanding beyond text to proxies learned from other modalities (see Knox et al. (2022)), providing more guidance for setting $\gamma_{\mathrm{high}}$ in our odds ratio heuristic, extending our method to incorporate categorical $U, W, Z$, and using soft probabilistic outputs from the zero-shot classifiers.

## Acknowledgements

We are very grateful for feedback from Naoki Egami and Zach Wood-Doughty as well as helpful comments from anonymous reviewers from NeurIPS 2024. KK acknowledges support from the Allen Institute for Artificial Intelligence's Young Investigator Grant. RB acknowledges support from the NSF CRII grant 2348287. The content of the information does not necessarily reflect the position or the policy of the Government, and no official endorsement should be inferred.

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

## A    Elaboration on Problem Restriction

Restriction (R1) is clearly present in settings for which we need to adjust for a confouding variable that is impossible or difficult to measure—for example, atrial fibrillation can go undiagnosed for years. A logical first attempt to mitigate this constraint is to train a supervised classifier on a subset of the data to create proxies for the rest of the dataset, as explored in Wood-Doughty et al. (2018). This, however, requires humans to hand-label large amounts of text data. If labeling takes place on a crowd-sourcing platform, e.g., Amazon's Mechanical Turk, crowd-sourcing costs can quickly sky-rocket and often exceed tens of thousands of dollars, even for small datasets. Furthermore, many datasets—particularly those in clinical settings—require domain expertise (e.g., trained medical doctors) which will likely increase costs significantly and limit the availability of labelers.

Cost and expertise of labelers aside, the possibility of supervised learning is further restricted by patient privacy legislation. We typically cannot transport sensitive data regarding patients' personal data to platforms such as Amazon's Mechanical Turk for labeling due to the Health Insurance Portability and Accountability Act (HIPAA)[12] in the United States. In addition, legal acts such as the General Data Protection Regulation (GDPR)[13] in the European Union and the California Consumer Privacy Act (CCPA)[14] restrict the movement and repurposing of user data across platforms. Our proposed method overcomes restriction (R1) by using zero-shot learners that do not require previously labeled examples.

## B    Using Post-Treatment or Actionable Text Data

Here, we first describe a scenario for which we may infer valid proxies using both post-treatment and pre-treatment text. In Figure 5, $\mathbf{T}^{\text{post}}$ is post-treatment text whereas $\mathbf{T}^{\text{pre}}$ is pre-treatment text. Through simple d-separation arguments, we can see that each of (P1-P3) are fulfilled. Hence, it is still possible to use post-treatment text to generate valid proxies as long as we use pre-treatment text to generate one proxy and post-treatment text to generate the other. However, requiring both instances of text data to be pre-treatment is simpler to implement and validate, so we recommend this rule in our final method.

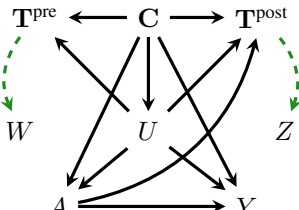

Figure 5: Using both pre-treatment and post-treatment text to generate valid proxies.

Figure 6 further illustrates an example where we can use actionable text to generate valid proxies. Suppose $\mathbf{T}^{\text{act}}$ is an instance of text data that influences a health practitioner's decision to assign different treatments to a patient, and $\mathbf{T}^{\text{pre}}$ is an instance of pre-treatment text that is not actionable. Then, $\mathbf{T}^{\text{act}}$ has a direct edge to the treatment $A$. Despite the addition of this edge, we can apply d-separation again to verify that (P1-P3) are still fulfilled. Therefore, our proposed method is still valid in a more relaxed setting where clinicians use one instance of the text data to make treatment decisions as long as the other instance of text data is both pre-treatment and not actionable.

---

[12]https://www.hhs.gov/hipaa/index.html
[13]https://www.consilium.europa.eu/en/policies/data-protection/
data-protection-regulation/
[14]https://oag.ca.gov/privacy/ccpa

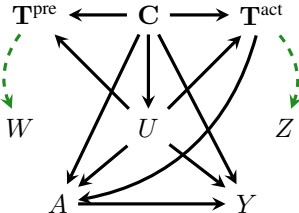

Figure 6: Using one instance of actionable text data and another instance of non-actionable text data to generate valid proxies.

## C Fully Synthetic Data-Generating Process

The DAG representing the fully synthetic data-generating process is shown in Figure 7. We simulate $U$, a binary variable, as follows

$$U \sim \text{Bernoulli}(0.48)$$

Next, we simulate baseline confounders and synthetic "text" $X_1, X_2, X_3, X_4$ as follows:

$$C \sim \mathcal{N}(0,1)$$
$$X_1 \sim \mathcal{N}(0,1) + 1.95 * U + 3 * C$$
$$X_2 \sim \mathcal{N}(0,1) + \exp(X_1) + U + 3 * C$$
$$X_3 \sim \mathcal{N}(0,1) + 1.25 * U + 3 * C$$
$$X_4 \sim \mathcal{N}(0,1) + X_3{}^2 + 0.5 * X_3{}^3$$
$$+ U + 3 * C$$

Finally, the treatment and outcome variables are generated via

$$p(A = 1) = \text{expit}(0.8 * U + C - 0.3)$$
$$A \sim \text{Bernoulli}(p(A = 1))$$
$$Y \sim \mathcal{N}(0,1) + 1.3 * A + 0.8 * U + C$$

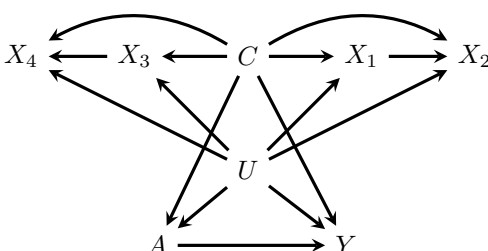

Figure 7: DAG showing the fully synthetic data-generating process.

In the fully synthetic data-generating process, increasing the coefficient for the variable $C$ decreases the odds ratio conditional on observed confounders $\gamma_{WZ.\mathbf{C}}$ while decreasing the coefficient for the variable $C$ increases $\gamma_{WZ.\mathbf{C}}$.

## D MIMIC-III Pre-Processing Steps

In the MIMIC-III dataset (Johnson et al., 2016), data is organized into multiple tables. Patients are anonymized and each have a unique identifier. In addition, each unique hospital admission is assigned a hospital admission identification number (HAID). Each patient can have multiple HAIDs, and each HAID can have multiple notes (e.g., an ECG note and a nursing note). Our pre-processing steps are outlined in the following steps:

1. We start with the table in MIMIC-III containing the clinicians' notes for all patients and drop rows of data where the HAID has a missing value. We refer to this main table we are working with as our **reference** table.

2. Next, for each unique patient in the **reference** table, we select one HAID by choosing the HAID with the earliest "chart date," the day on which the clinician's note was recorded. We limit each patient to only one hospital admission to better fit the assumption of i.i.d. data (which would be violated if there were multiple hospital admissions for a single patient since these multiple admissions would be dependent).

3. We drop all clinicians' notes that have the category label "Discharge summary" because such notes would be considered post-treatment text that violate (P2) as we have shown in Section 3. For each HAID, we combine separate notes of the same category into one string.

4. We now move on to the table in MIMIC-III that contains information on which patients were given which medical diagnoses for each HAID. For each HAID in the **reference** table, we select all the diagnoses (ICD-9 codes) assigned to the patient in that hospital visit.

5. The 10 most common diagnoses for the patients in our **reference** table are, in order, hypertension, coronary atherosclerosis of the native coronary artery, atrial fibrillation, congestive heart failure, diabetes mellitus, hyperlipidemia, acute kidney failure, need for vaccination against viral hepatitis, suspected newborn infection, and acute respiratory failure. In our **reference** table, we add a column for each of these 10 diagnoses and record a 0 or a 1 that indicates whether each patient was assigned a diagnosis for each condition on a particular hospital admission.

6. From the table containing baseline information for each patient, we append onto the **reference** table the gender and age of each patient. We infer each patient's age by subtracting the "chart date" of the patient's clinician's note by the patient's date of birth. We drop from the **reference** table all patients with an age less than 18 or greater than 100. We drop the diagnosis *suspected newborn infection* from the **reference** table because we have eliminated all patients younger than 18. At this stage in the preprocessing pipeline, the total number of HAIDs, or rows of data, in the **reference** table is 990,172.

7. In the final step, we evaluate which pairs of note categories are the most often simultaneously recorded for patients on the same hospital admission in the **reference** table. The top 5 simultaneously recorded note category pairs are Electrocardiogram (ECG) and Radiology with 27,963 HAIDs, Nursing and ECG with 18,505 HAIDs, Nursing and Radiology with 18,302 HAIDs, ECG and Echocardiogram (Echo) with 17,291 HAIDs, and Echo and Radiology with 15,558 HAIDs. We consider these 5 pairs of note categories for downstream proximal causal inference.

## E  Diagnosing Signal With Oracle Text Classifiers

To ensure that the text contains predictive signal for the diagnoses (a precondition necessary to use our zero-shot classifiers later on in our analysis), we train supervised classifiers to predict the diagnoses, $Y$, separately using text data from each note category, $X$, with a bag-of-words representation.

Our procedure is as follows. We first truncate all of the clinicians' notes data to 470 tokens (due to the context window of Flan-T5 which we use later in our pipeline). We ignore oracles and note categories combinations where positivity rates are 0 or 1 after subsetting on the note category in question. We then use the scikit-learn library's `CountVectorizer` to convert the text data to bag of words features with vocabulary size $5,000$ and train a linear logistic regression with the hyperparameter `penalty` set to "None". We further calculate the F1 score, accuracy, precision, and recall with the scikit-learn library Pedregosa et al. (2011). The results are summarized in Table 3.

We choose oracles with at least two note categories that achieve F1 scores greater than 0.7 for downstream inference with zero-shot classifiers. Such oracles and note categories are:

1. Atrial fibrillation (A-fib) with ECG, Echo, and Nursing notes.

2. Congestive heart failure (Heart) with Echo and Nursing notes.

3. Coronary atherosclerosis of the native coronary artery (A-sis) with Echo, Nursing, and Radiology notes.

4. Hypertension with Echo and Nursing notes.

| Diagnosis (Possible $U$) | Note Category | $p(U=1)$ | F1 Score | Accuracy | Precision | Recall |
|---|---|---|---|---|---|---|
| Hyperlipidemia | Nursing | 0.087 | 1.000 | 1.000 | 1.000 | 1.000 |
| Acute respiratory failure | Nursing | 0.132 | 1.000 | 1.000 | 1.000 | 1.000 |
| **Coronary atherosclerosis (A-sis)** | Nursing | 0.279 | 0.993 | 0.996 | 0.995 | 0.991 |
| Acute kidney failure | Nursing | 0.135 | 0.958 | 0.989 | 0.981 | 0.937 |
| **Atrial fibrillation (A-fib)** | Nursing | 0.231 | 0.879 | 0.946 | 0.909 | 0.851 |
| **Coronary atherosclerosis (A-sis)** | Echo | 0.375 | 0.844 | 0.888 | 0.885 | 0.807 |
| **Coronary atherosclerosis (A-sis)** | Radiology | 0.249 | 0.843 | 0.926 | 0.901 | 0.792 |
| **Congestive heart failure (Heart)** | Nursing | 0.225 | 0.789 | 0.910 | 0.840 | 0.744 |
| **Hypertension** | Nursing | 0.417 | 0.762 | 0.806 | 0.779 | 0.747 |
| **Congestive heart failure (Heart)** | Echo | 0.326 | 0.759 | 0.853 | 0.813 | 0.713 |
| **Atrial fibrillation (A-fib)** | Echo | 0.328 | 0.750 | 0.850 | 0.827 | 0.686 |
| Diabetes | Nursing | 0.168 | 0.744 | 0.922 | 0.828 | 0.675 |
| **Atrial fibrillation (A-fib)** | ECG | 0.263 | 0.742 | 0.883 | 0.886 | 0.638 |
| **Hypertension** | Echo | 0.457 | 0.712 | 0.744 | 0.733 | 0.693 |
| Hypertension | Radiology | 0.420 | 0.679 | 0.741 | 0.707 | 0.653 |
| Acute respiratory failure | Radiology | 0.145 | 0.675 | 0.917 | 0.781 | 0.594 |
| Hyperlipidemia | Echo | 0.226 | 0.662 | 0.866 | 0.768 | 0.581 |
| Congestive heart failure | Radiology | 0.207 | 0.661 | 0.877 | 0.771 | 0.578 |
| Atrial fibrillation | Radiology | 0.233 | 0.655 | 0.858 | 0.761 | 0.574 |
| Acute respiratory failure | Echo | 0.179 | 0.622 | 0.887 | 0.777 | 0.519 |
| Acute kidney failure | Radiology | 0.154 | 0.607 | 0.898 | 0.748 | 0.511 |
| Coronary atherosclerosis | ECG | 0.297 | 0.590 | 0.806 | 0.791 | 0.471 |
| Hyperlipidemia | Radiology | 0.163 | 0.578 | 0.885 | 0.723 | 0.481 |
| Acute kidney failure | Echo | 0.194 | 0.571 | 0.863 | 0.732 | 0.468 |
| Congestive heart failure | ECG | 0.238 | 0.555 | 0.835 | 0.772 | 0.433 |
| Hypertension | ECG | 0.445 | 0.541 | 0.658 | 0.672 | 0.453 |
| Diabetes | Echo | 0.201 | 0.524 | 0.851 | 0.729 | 0.409 |
| Diabetes | Radiology | 0.172 | 0.453 | 0.860 | 0.688 | 0.338 |
| Acute respiratory failure | ECG | 0.148 | 0.391 | 0.879 | 0.769 | 0.262 |
| Hyperlipidemia | ECG | 0.182 | 0.381 | 0.849 | 0.746 | 0.256 |
| Acute kidney failure | ECG | 0.167 | 0.365 | 0.861 | 0.774 | 0.239 |
| Diabetes | ECG | 0.187 | 0.307 | 0.839 | 0.785 | 0.191 |

Table 3: Descending order of F1 scores (as well as accuracy, precision and recall) from bag of words linear logistic regressions predicting various medical diagnoses from the MIMIC-III dataset using different medical note categories as input. We set $0.7$ as our F1 score cutoff for choosing the diagnoses to use in downstream analysis with zero-shot classifiers (dashed horizontal line). We bold the diagnoses that passed this threshold and also have at least two note categories. We used these (bolded) diagnoses in the final oracle $U$ in downstream analyses.

## F   LLM and Estimation Details

**Choosing LLMs: FLAN and OLMo**   We used the large language models FLAN-T5 XXL and OLMo-7B-Instruct for inferring the oracle diagnoses from text data Chung et al. (2024); Groeneveld et al. (2024). These are both fully "open" models in that we have full knowledge of their pre-training data, training objectives, and training details, and can be downloaded and updated on a local machine, all of which are important for reproducible science (Palmer et al., 2024). We chose these two models because they have been fine-tuned for zero-shot classification and because they have been trained on different text corpora. Furthermore, we hypothesize that Flan will perform well on clinical notes because, according to Dodge et al. (2021)'s analysis, C4 (the pre-training data in Flan-T5) has among its top 25 domains `patents.google.com`, `journals.plos.org`, `link.springer.com`, `www.ncbi.nlm.nih.gov`, all of which likely contain medical text.

**Creating inferences from FLAN and OLMo**   We use the following string as the prompt for both models: '*Context: {$\mathbf{T}^{pre}$} \nIs it likely the patient has {$U$}?\nConstraint: Even if you are uncertain, you must pick either "Yes" or "No" without using any other words.*' We consider only the first five tokens of output for both models and assign a value of $1$ for the proxy if the output string converted to all lowercase characters contains the word '*yes*' and 0 otherwise.

**Odds ratio confidence interval estimation**  We calculate a point estimate for the odds ratio $\gamma_{WZ.\mathbf{C}}$ using the scikit-learn library (Pedregosa et al., 2011) by first training a linear logistic regression with $W$ as the target outcome and $\{Z\} \cup \mathbf{C}$ as the features. Whenever the positivity rate of $W$ is less than $0.2$ or greater than $0.8$, i.e. there is a class imbalance, we set the hyperparameter `class_weight` to "`balanced`". This uses the values of $W$ to automatically adjust weights inversely proportional to class frequencies in the dataset. Regardless of whether we adjust for class imbalance, we set the hyperparameter `penalty` to None to turn off regularization. Finally, we calculate the natural exponent of the coefficient of $Z$ as the point estimate for the odds ratio $\gamma_{WZ.\mathbf{C}}$.

We calculate the confidence interval for the odds ratio by drawing 200 bootstrap samples and repeating the steps above for each bootstrap sample to create a bootstrap distribution. We then take the 0.025th and 0.975th percentiles of the bootstrap distribution as $\gamma_{WZ.\mathbf{C}}^{\text{CI low}}$ and $\gamma_{WZ.\mathbf{C}}^{\text{CI low}}$, respectively.

**Two-stage linear proximal causal inference estimator for the ACE**  Following sample splitting from the causal inference literature Hansen (2000), we start by splitting the semi-synthetic dataset into two splits—split 1 and split 2—where both splits are $50\%$ of the original dataset. We then train a linear logistic regression using split 1 with $W$ as the target outcome and $\{A, Z\} \cup \mathbf{C}$ as the features. Like in the estimation procedure for the odds ratio, we set the hyperparameter `class_weight` to "`balanced`" whenever there is a class imbalance in the target outcome $W$ and always set the hyperparameter `penalty` to None. Next, we make probabilistic predictions $\widehat{W}$ for $W$ with the previously trained linear logistic regression model using split 2. Finally, we train a linear regression using split 2 of the data with $Y$ as the target outcome and $\{A, \widehat{W}\} \cup \mathbf{C}$ as the features. The coefficient of $A$ in the trained linear regression model is the estimate for the ACE.

We calculate the confidence interval for the ACE by drawing 200 bootstrap samples and repeating the steps above for each bootstrap sample to create a bootstrap distribution. We then take the 0.025th and 0.975th percentiles of the bootstrap distribution as the lower and upper bound of the confidence interval for the ACE, respectively.

## G  Semi-Synthetic Data Generating Process

The DAG representing the semi-synthetic data-generating process is shown in Figure 8. The variable $U$ represents `atrial fibrillation`, `congestive heart failure`, `coronary atherosclerosis of the native coronary artery`, or `hypertension`, depending on the setting. We use the **reference** table created in the pre-processing steps described in Section D of the Appendix and start by normalizing the variable Age with the formula

$$\text{Age}_i = \frac{\text{Age}_i - \bar{X}_{\text{Age}}}{\sigma_{\text{Age}}}.$$

We then simulate draws of the binary variable $A$ via

$$p(A = 1) = \text{expit}(U + 0.9 * \text{Gender}$$
$$+ 0.9 * \text{Age})$$
$$A \sim \text{Bernoulli}(p(A = 1))$$

Next, we simulate draws of the continuous variable $Y$ from:

$$Y \sim \mathcal{N}(0, 1) + 1.3 * A + U$$
$$+ 0.9 * \text{Gender} + 0.9 * \text{Age}$$

When estimating the ACE using the two-stage linear regression described in Section 5, we condition on the baseline covariates Age and Gender as well the other diagnoses not being used as $U$.

## H  Additional P2M Experiments

In Figure 9, we report additional experimental results from P1M using Flan to infer proxies from both note categories and P2M using Flan and OLMo to infer proxies from one note category each. As expected, using one inferred proxy from an LLM directly in backdoor adjustment produces biased estimates for the ACE. When using P1M and P2M, the odds ratio falsification heuristic successfully

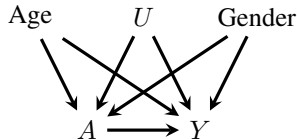

Figure 8: Causal DAG showing the semi-synthetic data-generating process.

flags invalid estimates of the ACE. In the two cases where the inferred proxies pass the odds ratio falsification heuristic, we observe valid estimates for the ACE.

A notable example of when the odds ratio heuristic correctly failed is for Echo and Nursing notes on the diagnosis hypertension. As shown in Figure 9, the estimates for the ACE for both P1M and P2M are far from the true ACE of 1.3. This is explained by Table 21 where the oracle odds ratios between both proxies $W$ and $Z$ and the oracle $U$ are nearly 1, suggesting that the zero-shot classifiers were not able to accurately predict the oracle. As expected in such a scenario, the estimates for the ACE are biased, and our odds ratio heuristic successfully detects the violation of Assumption (P4) for proximal causal inference.

## I  Examining Text Conditional Independence

Recall, our procedure requires us to split text such that $\mathbf{T}_1^{\text{pre}} \not\perp\!\!\!\perp \mathbf{T}_2^{\text{pre}} \mid U, \mathbf{C}$. This condition must hold at inference time when practioners are using our procedure and it must hold in the "real" part of our semi-synthetic experiments in Section 5. Thus, for our semi-synthetic experiments with real-world clinical notes, we aim to investigate when this condition does or does not hold. To the best of our knowledge, it is an open problem on how to perform (conditional) independence tests on high-dimensional, compositional language data. Instead, we turn to an approximation that can lend us some preliminary evidence and examine the probabilities of top unigram features for each pairing of text categories. We show the results in Tables 4 and 5 below.

After tokenization into unigrams, we select the top five features via tf-idf scores. The tf-idf scores are created under two settings: (1) fitting tf-idf on both note categories **combined**, or (2) fitting tf-idf separately on each note category. Unique features (unigrams) must occur in a minimum of 10% of the documents and a maximum of 90% of the documents to be considered. Note, when we look at combinations of note pairings we subset to patients that have *both* notes, so subsets of documents are overlapping but not the exact same, resulting in different top features for the same category in different note pairs.

## J  Key Metrics from Semi-Synthetic Simulations

We report key metrics, mostly requiring access to the oracle $U$, for the semi-synthetic simulations in the tables below. Although these metrics will typically not be available to practitioners at inference time, they are helpful in understanding the relationship between the inferred proxies $W$ and $Z$ under various conditions. The odds ratio values between the proxies and the oracle, $\gamma_{WU.\mathbf{C}}$ and $\gamma_{ZU.\mathbf{C}}$, show how correlated the proxies are with the oracle, an important indicator for (P4). The difference between 1 and the positivity rate of the oracle describes the minimum accuracy threshold for which the inferred proxy must perform to be more accurate than simply predicting the majority class.

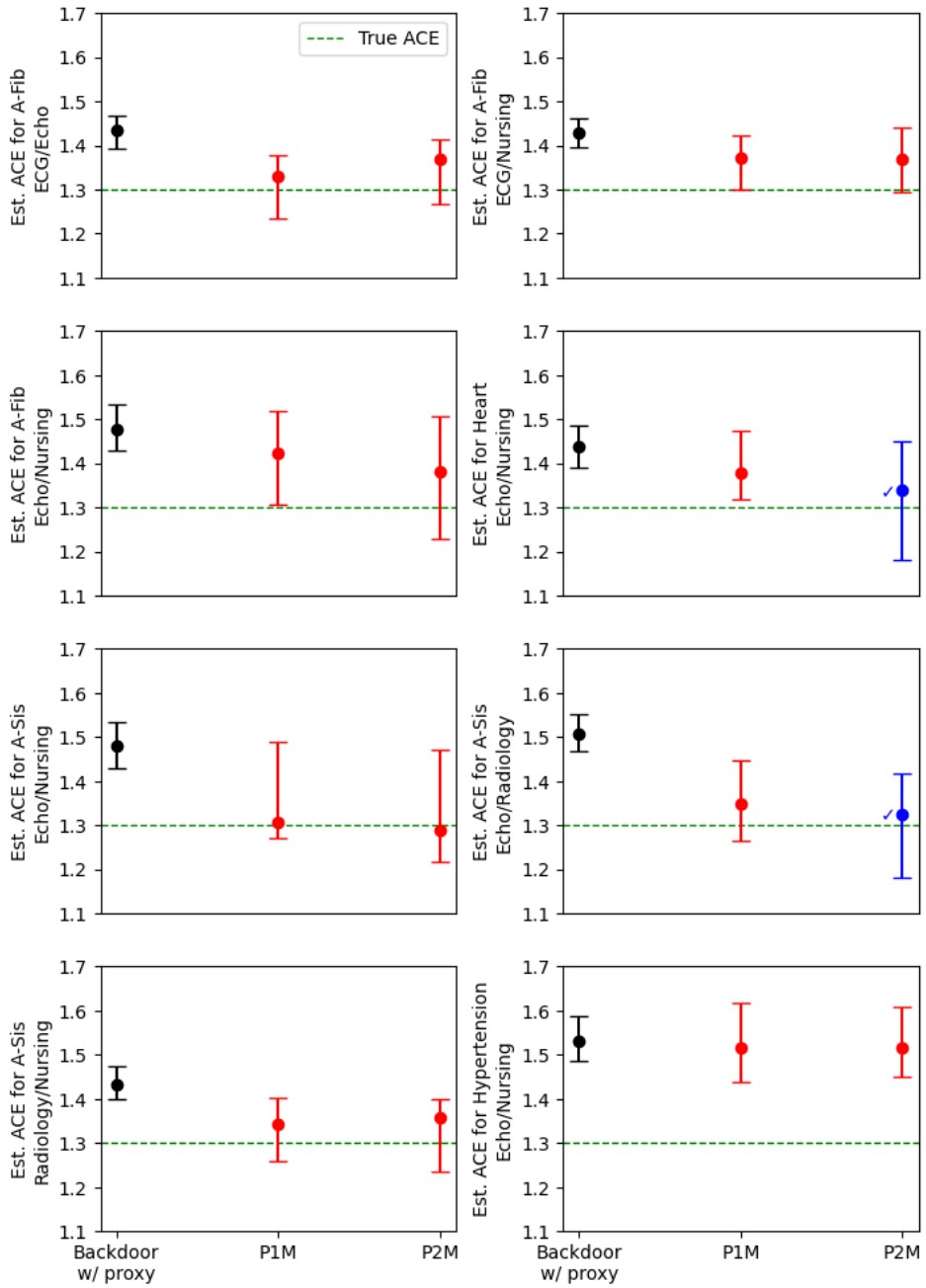

Figure 9: **Semi-synthetic results** for ACE point estimates (dots) and confidence intervals (bars) for all oracles and note categories that we detected sufficient signal in. Here, ✓ distinguishes settings that passed the odds ratio heuristic from those that failed it, with $\gamma_{\text{high}} = 2$.

|  | aorta | mild | mildly | rhythm | thickened |
|---|---|---|---|---|---|
| ECG | 0.000 | 0.026 | 0.003 | 0.851 | 0.000 |
| Echo | 0.897 | 0.847 | 0.845 | 0.150 | 0.743 |
|  | hr | name | rhythm | st | the |
| ECG | 0.000 | 0.004 | 0.824 | 0.581 | 0.638 |
| Nursing | 0.872 | 0.816 | 0.113 | 0.266 | 0.647 |
|  | chest | contrast | left | pm | right |
| ECG | 0.004 | 0.000 | 0.385 | 0.000 | 0.175 |
| Radiology | 0.845 | 0.517 | 0.824 | 0.876 | 0.858 |
|  | mild | mildly | name | thickened | was |
| Echo | 0.830 | 0.845 | 0.137 | 0.734 | 0.388 |
| Nursing | 0.094 | 0.018 | 0.810 | 0.004 | 0.596 |
|  | chest | dilated | mild | mildly | thickened |
| Echo | 0.072 | 0.695 | 0.839 | 0.838 | 0.732 |
| Radiology | 0.883 | 0.071 | 0.420 | 0.102 | 0.018 |
|  | chest | ct | left | right | was |
| Nursing | 0.306 | 0.521 | 0.389 | 0.339 | 0.600 |
| Radiology | 0.834 | 0.556 | 0.827 | 0.856 | 0.417 |

Table 4: **Combined tf-idf.** For each pair of note categories (subsequent rows), we selected the top five unigram features (columns) via tf-idf scores on the combined note pairs. We report the proportion of notes in each category that contain that feature.

| ECG | [is, of, rhythm, the, wave] |
|---|---|
| Echo | [aorta, mild, mildly, of, thickened] |
| ECG | [of, rhythm, st, the, wave] |
| Nursing | [hr, is, name, this, was] |
| ECG | [is, of, rhythm, the, wave] |
| Radiology | [chest, contrast, left, right, to] |
| Echo | [dilated, mild, mildly, thickened, with] |
| Nursing | [hr, is, name, this, was] |
| Echo | [aorta, mild, mildly, of, thickened] |
| Radiology | [chest, contrast, left, pm, right] |
| Nursing | [hr, is, name, neuro, this] |
| Radiology | [chest, contrast, left, right, to] |

Table 5: **Separate tf-idf.** For each pair of note categories (subsequent rows), we report the top five unigram features via tf-idf scores that were fit on each note category separately.

| Oracle? | Proxies | Metric | $U$=A-Fib |
|---|---|---|---|
| Yes | – | 1 - $p(U = 1)$ | 0.665 |
| Yes | $W$ from Flan on $\mathbf{T}_1^{\text{pre}}$ (ECG) | $\gamma_{WU.\mathbf{C}}$
Accuracy
$p(W = 1)$
Precision
Recall | 21.255
0.802
0.186
0.870
0.481 |
| Yes | $Z$ from Flan on $\mathbf{T}_1^{\text{pre}}$ (Echo) | $\gamma_{ZU.\mathbf{C}}$
Accuracy
$p(Z = 1)$
Precision
Recall | 17.815
0.716
0.064
0.899
0.172 |
| No | $Z, W$ | Raw Agreement Rate; $p(W = Z)$ | 0.841 |

Table 6: Key metrics for the diagnosis atrial fibrillation when inferring proxies from ECG and Echo clinicians' notes with one zero-shot classifier.

| Oracle? | Proxies | Metric | $U$=A-Fib |
|---|---|---|---|
| Yes | – | 1 - $p(U = 1)$ | 0.665 |
| Yes | $W$ from Flan on $\mathbf{T}_1^{\text{pre}}$ (ECG) | $\gamma_{WU.\mathbf{C}}$
Accuracy
$p(W = 1)$
Precision
Recall | 21.255
0.802
0.186
0.870
0.481 |
| Yes | $Z$ from OLMo on $\mathbf{T}_1^{\text{pre}}$ (Echo) | $\gamma_{ZU.\mathbf{C}}$
Accuracy
$p(Z = 1)$
Precision
Recall | 2.666
0.677
0.170
0.537
0.272 |
| No | $Z, W$ | Raw Agreement Rate; $p(W = Z)$ | 0.766 |

Table 7: Key metrics for the diagnosis atrial fibrillation when inferring proxies from ECG and Echo clinicians' notes with two zero-shot classifiers.

| Oracle? | Proxies | Metric | $U$=A-Fib |
|---|---|---|---|
| Yes | – | 1 - $p(U = 1)$ | 0.738 |
| Yes | $W$ from Flan on $\mathbf{T}_1^{\text{pre}}$ (ECG) | $\gamma_{WU.\mathbf{C}}$
Accuracy
$p(W = 1)$
Precision
Recall | 23.825
0.839
0.143
0.853
0.466 |
| Yes | $Z$ from Flan on $\mathbf{T}_1^{\text{pre}}$ (Nursing) | $\gamma_{ZU.\mathbf{C}}$
Accuracy
$p(Z = 1)$
Precision
Recall | 6.751
0.794
0.240
0.615
0.563 |
| No | $Z, W$ | Raw Agreement Rate; $p(W = Z)$ | 0.812 |

Table 8: Key metrics for the diagnosis atrial fibrillation when inferring proxies from ECG and Nursing clinicians' notes with one zero-shot classifier.

| Oracle? | Proxies | Metric | $U$=A-Fib |
|---|---|---|---|
| Yes | – | 1 - $p(U = 1)$ | 0.738 |
| Yes | $W$ from Flan on $\mathbf{T}_1^{\text{pre}}$ (ECG) | $\gamma_{WU.\mathbf{C}}$
Accuracy
$p(W = 1)$
Precision
Recall | 23.825
0.839
0.143
0.853
0.466 |
| Yes | $Z$ from OLMo on $\mathbf{T}_1^{\text{pre}}$ (Nursing) | $\gamma_{ZU.\mathbf{C}}$
Accuracy
$p(Z = 1)$
Precision
Recall | 2.400
0.689
0.288
0.415
0.456 |
| No | $Z, W$ | Raw Agreement Rate; $p(W = Z)$ | 0.716 |

Table 9: Key metrics for the diagnosis atrial fibrillation when inferring proxies from ECG and Nursing clinicians' notes with two zero-shot classifiers.

| Oracle? | Proxies | Metric | $U$=A-Fib |
|---------|---------|--------|-----------|
| Yes | – | $1 - p(U=1)$ | 0.685 |
| Yes | $W$ from Flan on $\mathbf{T}_1^{\text{pre}}$ (Echo) | $\gamma_{WU.\mathbf{C}}$
Accuracy
$p(W=1)$
Precision
Recall | 15.959
0.731
0.060
0.885
0.169 |
| Yes | $Z$ from Flan on $\mathbf{T}_1^{\text{pre}}$ (Nursing) | $\gamma_{ZU.\mathbf{C}}$
Accuracy
$p(Z=1)$
Precision
Recall | 6.044
0.760
0.283
0.632
0.569 |
| No | $Z, W$ | Raw Agreement Rate; $p(W=Z)$ | 0.746 |

Table 10: Key metrics for the diagnosis atrial fibrillation when inferring proxies from Echo and Nursing clinicians' notes with one zero-shot classifier.

| Oracle? | Proxies | Metric | $U$=A-Fib |
|---------|---------|--------|-----------|
| Yes | – | $1 - p(U=1)$ | 0.685 |
| Yes | $W$ from Flan on $\mathbf{T}_1^{\text{pre}}$ (Echo) | $\gamma_{WU.\mathbf{C}}$
Accuracy
$p(W=1)$
Precision
Recall | 15.959
0.731
0.060
0.885
0.169 |
| Yes | $Z$ from OLMo on $\mathbf{T}_1^{\text{pre}}$ (Nursing) | $\gamma_{ZU.\mathbf{C}}$
Accuracy
$p(Z=1)$
Precision
Recall | 2.237
0.655
0.318
0.454
0.458 |
| No | $Z, W$ | Raw Agreement Rate; $p(W=Z)$ | 0.683 |

Table 11: Key metrics for the diagnosis atrial fibrillation when inferring proxies from Echo and Nursing clinicians' notes with two zero-shot classifiers.

| Oracle? | Proxies | Metric | $U$=Heart |
|---------|---------|--------|-----------|
| Yes | – | $1 - p(U=1)$ | 0.651 |
| Yes | $W$ from Flan on $\mathbf{T}_1^{\text{pre}}$ (Echo) | $\gamma_{WU.\mathbf{C}}$
Accuracy
$p(W=1)$
Precision
Recall | 3.863
0.704
0.297
0.591
0.502 |
| Yes | $Z$ from Flan on $\mathbf{T}_1^{\text{pre}}$ (Nursing) | $\gamma_{ZU.\mathbf{C}}$
Accuracy
$p(Z=1)$
Precision
Recall | 2.450
0.643
0.407
0.490
0.572 |
| No | $Z, W$ | Raw Agreement Rate; $p(W=Z)$ | 0.648 |

Table 12: **This setting was included in the main text of the paper.** Key metrics for the diagnosis congestive heart failure when inferring proxies from Echo and Nursing clinicians' notes with one zero-shot classifier.

| Oracle? | Proxies | Metric | $U$=Heart |
|---|---|---|---|
| Yes | – | $1 - p(U = 1)$ | 0.651 |
| Yes | $W$ from Flan on $\mathbf{T}_1^{\text{pre}}$ (Echo) | $\gamma_{WU.\mathbf{C}}$
Accuracy
$p(W = 1)$
Precision
Recall | 3.863
0.704
0.297
0.591
0.502 |
| Yes | $Z$ from OLMo on $\mathbf{T}_1^{\text{pre}}$ (Nursing) | $\gamma_{ZU.\mathbf{C}}$
Accuracy
$p(Z = 1)$
Precision
Recall | 1.416
0.588
0.357
0.412
0.421 |
| No | $Z, W$ | Raw Agreement Rate; $p(W = Z)$ | 0.590 |

Table 13: **This setting was included in the main text of the paper.** Key metrics for the diagnosis congestive heart failure when inferring proxies from Echo and Nursing clinicians' notes with two zero-shot classifiers.

| Oracle? | Proxies | Metric | $U$=A-Sis |
|---|---|---|---|
| Yes | – | $1 - p(U = 1)$ | 0.627 |
| Yes | $W$ from Flan on $\mathbf{T}_1^{\text{pre}}$ (Echo) | $\gamma_{WU.\mathbf{C}}$
Accuracy
$p(W = 1)$
Precision
Recall | 6.232
0.704
0.156
0.746
0.313 |
| Yes | $Z$ from Flan on $\mathbf{T}_1^{\text{pre}}$ (Nursing) | $\gamma_{ZU.\mathbf{C}}$
Accuracy
$p(Z = 1)$
Precision
Recall | 10.546
0.785
0.331
0.739
0.656 |
| No | $Z, W$ | Raw Agreement Rate; $p(W = Z)$ | 0.721 |

Table 14: Key metrics for the diagnosis coronary atherosclerosis when inferring proxies from Echo and Nursing clinicians' notes with one zero-shot classifier.

| Oracle? | Proxies | Metric | $U$=A-Sis |
|---|---|---|---|
| Yes | – | $1 - p(U = 1)$ | 0.627 |
| Yes | $W$ from Flan on $\mathbf{T}_1^{\text{pre}}$ (Echo) | $\gamma_{WU.\mathbf{C}}$
Accuracy
$p(W = 1)$
Precision
Recall | 6.232
0.704
0.156
0.746
0.313 |
| Yes | $Z$ from OLMo on $\mathbf{T}_1^{\text{pre}}$ (Nursing) | $\gamma_{ZU.\mathbf{C}}$
Accuracy
$p(Z = 1)$
Precision
Recall | 4.134
0.680
0.135
0.693
0.251 |
| No | $Z, W$ | Raw Agreement Rate; $p(W = Z)$ | 0.787 |

Table 15: Key metrics for the diagnosis coronary atherosclerosis when inferring proxies from Echo and Nursing clinicians' notes with two zero-shot classifiers.

| Oracle? | Proxies | Metric | $U$=A-Sis |
|---|---|---|---|
| Yes | – | 1 - $p(U = 1)$ | 0.643 |
| Yes | $W$ from Flan on $\mathbf{T}_1^{\text{pre}}$ (Echo) | $\gamma_{WU.\mathbf{C}}$
Accuracy
$p(W = 1)$
Precision
Recall | 6.240
0.708
0.133
0.745
0.277 |
| Yes | $Z$ from Flan on $\mathbf{T}_1^{\text{pre}}$ (Radiology) | $\gamma_{ZU.\mathbf{C}}$
Accuracy
$p(Z = 1)$
Precision
Recall | 6.344
0.727
0.180
0.733
0.370 |
| No | $Z, W$ | Raw Agreement Rate; $p(W = Z)$ | 0.784 |

Table 16: **This setting was included in the main text of the paper.** Key metrics for the diagnosis coronary atherosclerosis when inferring proxies from Echo and Radiology clinicians' notes with one zero-shot classifier.

| Oracle? | Proxies | Metric | $U$=A-Sis |
|---|---|---|---|
| Yes | – | 1 - $p(U = 1)$ | 0.643 |
| Yes | $W$ from Flan on $\mathbf{T}_1^{\text{pre}}$ (Echo) | $\gamma_{WU.\mathbf{C}}$
Accuracy
$p(W = 1)$
Precision
Recall | 6.240
0.708
0.133
0.745
0.277 |
| Yes | $Z$ from OLMo on $\mathbf{T}_1^{\text{pre}}$ (Radiology) | $\gamma_{ZU.\mathbf{C}}$
Accuracy
$p(Z = 1)$
Precision
Recall | 5.046
0.678
0.069
0.753
0.146 |
| No | $Z, W$ | Raw Agreement Rate; $p(W = Z)$ | 0.830 |

Table 17: **This setting was included in the main text of the paper.** Key metrics for the diagnosis coronary atherosclerosis when inferring proxies from Echo and Radiology clinicians' notes with two zero-shot classifiers.

| Oracle? | Proxies | Metric | $U$=A-Sis |
|---|---|---|---|
| Yes | – | 1 - $p(U = 1)$ | 0.735 |
| Yes | $W$ from Flan on $\mathbf{T}_1^{\text{pre}}$ (Radiology) | $\gamma_{WU.\mathbf{C}}$
Accuracy
$p(W = 1)$
Precision
Recall | 7.265
0.784
0.129
0.690
0.335 |
| Yes | $Z$ from Flan on $\mathbf{T}_1^{\text{pre}}$ (Nursing) | $\gamma_{ZU.\mathbf{C}}$
Accuracy
$p(Z = 1)$
Precision
Recall | 12.800
0.833
0.224
0.719
0.606 |
| No | $Z, W$ | Raw Agreement Rate; $p(W = Z)$ | 0.793 |

Table 18: **This setting was included in the main text of the paper.** Key metrics for the diagnosis coronary atherosclerosis when inferring proxies from Radiology and Nursing clinicians' notes with one zero-shot classifier.

| Oracle? | Proxies | Metric | $U$=A-Sis |
|---|---|---|---|
| Yes | – | 1 - $p(U = 1)$ | 0.735 |
| Yes | $W$ from Flan on $\mathbf{T}_1^{\text{pre}}$ (Radiology) | $\gamma_{WU.\mathbf{C}}$
Accuracy
$p(W = 1)$
Precision
Recall | 7.265
0.784
0.129
0.690
0.335 |
| Yes | $Z$ from OLMo on $\mathbf{T}_1^{\text{pre}}$ (Nursing) | $\gamma_{ZU.\mathbf{C}}$
Accuracy
$p(Z = 1)$
Precision
Recall | 4.081
0.754
0.104
0.590
0.231 |
| No | $Z, W$ | Raw Agreement Rate; $p(W = Z)$ | 0.821 |

Table 19: **This setting was included in the main text of the paper.** Key metrics for the diagnosis coronary atherosclerosis when inferring proxies from Radiology and Nursing clinicians' notes with two zero-shot classifiers.

| Oracle? | Proxies | Metric | $U$=Hypertension |
|---|---|---|---|
| Yes | – | 1 - $p(U = 1)$ | 0.569 |
| Yes | $W$ from Flan on $\mathbf{T}_1^{\text{pre}}$ (Echo) | $\gamma_{WU.\mathbf{C}}$
Accuracy
$p(W = 1)$
Precision
Recall | 1.057
0.488
0.648
0.438
0.657 |
| Yes | $Z$ from Flan on $\mathbf{T}_1^{\text{pre}}$ (Nursing) | $\gamma_{ZU.\mathbf{C}}$
Accuracy
$p(Z = 1)$
Precision
Recall | 1.414
0.532
0.619
0.470
0.675 |
| No | $Z, W$ | Raw Agreement Rate; $p(W = Z)$ | 0.546 |

Table 20: Key metrics for the diagnosis hypertension when inferring proxies from Echo and Nursing clinicians' notes with one zero-shot classifier.

| Oracle? | Proxies | Metric | $U$=Hypertension |
|---|---|---|---|
| Yes | – | 1 - $p(U = 1)$ | 0.569 |
| Yes | $W$ from Flan on $\mathbf{T}_1^{\text{pre}}$ (Echo) | $\gamma_{WU.\mathbf{C}}$
Accuracy
$p(W = 1)$
Precision
Recall | 1.057
0.488
0.648
0.438
0.657 |
| Yes | $Z$ from OLMo on $\mathbf{T}_1^{\text{pre}}$ (Nursing) | $\gamma_{ZU.\mathbf{C}}$
Accuracy
$p(Z = 1)$
Precision
Recall | 1.053
0.524
0.432
0.448
0.449 |
| No | $Z, W$ | Raw Agreement Rate; $p(W = Z)$ | 0.494 |

Table 21: Key metrics for the diagnosis hypertension when inferring proxies from Echo and Nursing clinicians' notes with two zero-shot classifiers.

# K    Compute Resources

To run the experiments in this paper, we used a local server with 64 cores of CPUs and 4 x NVIDIA RTX A6000 48GB GPUs. Our semi-synthetic pipeline ran for approximately 36 hours where Flan took roughly 12 hours and OLMo took roughly 24 hours to compute inferences from the text data.

