# OpenReview forum: "Proximal Causal Inference With Text Data"
_NeurIPS.cc/2024/Conference — NeurIPS 2024 poster_

### Official Review · Reviewer_uHfh · 2024-07-03

**Soundness:** 4
**Presentation:** 3
**Contribution:** 4
**Rating:** 5
**Confidence:** 4

**Summary:**

The situation with unobserved confounding variables is quite common in applied research and makes causal inference complicated. To deal with such settings, the authors propose a new causal inference method that uses multiple instances of pre-treatment text data, estimates two proxies from two zero-shot models on the separate instances, and applies these proxies in the proximal g-formula. It is shown that the identification conditions are satisfied, The method is evaluated in a simulation study on synthetic and semi-synthetic data.

**Strengths:**

- The paper proposed a new innovative method for proximal causal inference utilizing text data.
- The paper is well written and motivated.

**Weaknesses:**

- Literature review is incomplete. The idea of proximal inference goes back to Zvi Griliches (1977) who should get credit for it. As background reading the corresponding chapter in the textbook "Applied Causal Inference Powered by ML and AI" by Chernozhukov and coauthors (2024).
- To overcome the completeness condition the paper "Causal Inference Under Unmeasured Confounding With Negative Controls: A Minimax Learning Approach" (arXiv:2103.14029) might be helpful.
- The theoretical results (Prop 1-4) are all in some sense "negative" and straightforward / already known. What would be more interesting, would be some "positive" results about the proposed method, although I am aware that this might be quite challenging.

**Questions:**

- How credible / realistic are (P1) and that $T^1_pre$ is conditionally independent from $T^2_pre$ in typical applications (e.g. using electronic health records)?
-  How could one use EHR to construct proxies? Are the assumptions fulfilled? (Might be questionable.. the example mentioned in the text is not convincing.)

**Limitations:**

Limitations are adressed adequately. As mentioned above, theoretical guarantees for the propsoed method might be very valuable.

---

> ### Author Rebuttal · Authors · 2024-08-06
>
> We thank the reviewer for the thoughtful feedback. We address the reviewer’s questions and comments on our paper's weaknesses.
>
> > Literature review is incomplete. The idea of proximal inference goes back to Zvi Griliches (1977) who should get credit for it. As background reading the corresponding chapter in the textbook "Applied Causal Inference Powered by ML and AI" by Chernozhukov and coauthors (2024).
>
> Thank you for pointing us to the paper. We are happy to point to it as a reference for causal inference with proxies in linear SEMs and that later works generalize this to non-linear settings.
>
> > To overcome the completeness condition the paper "Causal Inference Under Unmeasured Confounding With Negative Controls: A Minimax Learning Approach" (arXiv:2103.14029) might be helpful.
>
> Thank you for the reference. We believe this work complements our own as it concerns estimation given a pair of valid proxies. In our work, we provide a way of generating proxies from unstructured text that first satisfy the structural assumptions—assumptions P1-P3 that are encoded in the causal DAG Figure 1(b) of our paper and Figure 1 of the suggested reference—that are needed for both classical estimators in the proximal causal inference literature as well as the estimators developed in the suggested reference.
>
> The estimators in the reference depart from older proximal literature on assumption P4 of completeness. Importantly, however, the old and new estimators both require the $U \rightarrow Z$ and $U \rightarrow W$ edges to be real, i.e., the proxies must be at least weakly correlated with the unmeasured confounder. In our algorithm for generating proxies, when the zero-shot models generate even weakly predictive proxies, they automatically satisfy completeness and therefore the alternative assumptions suggested in the reference since in the discrete case the assumptions in the reference are strictly weaker (and so if completeness holds, these assumptions also hold). Thus, the estimators developed in the reference could be directly applied to the proxies generated by our algorithm. It would be interesting to see what would happen if one were to try and use zero-shot models to predict continuous proxies—in this case completeness and the assumptions listed in the reference are incomparable, which would necessitate the use of different estimators depending on what conditions were more likely to hold.
>
> We are happy to add a short discussion on this with the additional space granted in the camera ready if the paper is accepted.
>
> > The theoretical results (Prop 1-4) are all in some sense "negative" and straightforward / already known. What would be more interesting, would be some "positive" results about the proposed method, although I am aware that this might be quite challenging.
>
> We respectfully push back on this. Propositions 1-3 are indeed “negative” in that they correspond to results that show what cannot be done in order to obtain unbiased causal effect estimates. This is the same sense in which non-identification/completeness results in causal inference and missing data are negative results — they inform the reader about datasets/data collection procedures that would lead to biased estimates, e.g., [Shpitser and Pearl, 2006](https://ftp.cs.ucla.edu/pub/stat_ser/r327.pdf) and [Nabi et al, 2020](https://arxiv.org/pdf/2004.04872). Further, while Propositions 1-3 appear simple, they are not obvious: We have, for example, encountered [applied work](https://arxiv.org/pdf/2307.03687) that treats machine learned predictions as direct plug-ins for unmeasured variables—this is the same sense in which collider bias can be obvious, but only to those who know it and are already looking for it.
>
> Proposition 4 on the other hand is a “positive” result that shows identification and unbiased estimates are possible when following our algorithm. While these conditions may not always be fulfilled (just as with any other identification conditions) we also propose an odds ratio heuristic to check for violations of these conditions. The semi-synthetic results (Table 2 and Fig 3)  that use real-world text data from MIMIC-III also provide positive results (estimates with low bias) for the theory we propose.
>
> > How credible / realistic are (P1) and that $T^1_{pre}$ is conditionally independent from $T^2_{pre}$  in typical applications (e.g. using electronic health records)?
>
> We address this issue in our paper: In line 302-303, we write “We hypothesize that notes written by different individuals will satisfy $T^{pre}_1 \perp T^{pre}_2 | U, C$ since each individual will write a conditionally independent realization of 304 the patient’s status”. That is, this independence is more likely to hold if you use different clinical notes from different departments, in social media use different posts perhaps. We also propose the odds ratio heuristic as a way for the analyst to safeguard against violations of this condition.
>
> > How could one use EHR to construct proxies? Are the assumptions fulfilled? (Might be questionable.. the example mentioned in the text is not convincing.)
>
> We refer the reviewer to our semi-synthetic experiments in Section 5 that use real EHR data and clinical notes from MIMIC-III to do exactly this. Table 2 and Fig 3 shows the success of our proxy generation process as well as the success of the odds ratio heuristic as a safeguard when the assumptions are not fulfilled. We have also added an illustrative figure of our proxy generation process that can be found in the rebuttal PDF and that we would be happy to add to the camera ready if the paper is accepted.

---

> > ### Comment · Reviewer_uHfh · 2024-08-10
> > **Comment**
> >
> > First of all, thanks a lot for your efforts and the detailed answer. I still think that the negative results in Proposition 1-3 are not really deep and well known. I am aware that the reference can be considered as a complement, but I think a thorough literature review should contain all related sources. Finally, I still find the assumption of conditional independence hard to hold and I think here more convincing examples would be helpful. Overall, I will keep the score.

---

### Official Review · Reviewer_iLUt · 2024-07-11

**Soundness:** 3
**Presentation:** 3
**Contribution:** 3
**Rating:** 6
**Confidence:** 4

**Summary:**

Causal inference techniques often rely on the assumption that all confounders can be observed or inferred from available data. This paper proposes a method to address the setting where a confounder is entirely unobserved; instead there is available pre-treatment text that can be used to infer proxies, which in turn have predictive value for the unobserved confounder. The authors adapt proximal causal inference ideas to text-data settings, discuss "gotchas" in naive implementations of such a method, and explore various relevant choices before presenting best performing method. Specifically, they use multiple instances of pre-treatment text to infer two proxies for each datapoint using two different zero-shot language models. This is then used for average causal effect estimation via the proximal g-formula. The paper provides empirical evaluation on synthetic and semi-synthetic datasets as well as a falsifcation heuristic for untestable assumptions.

**Strengths:**

Originality: While proximal causal inference has been proposed and studied, this paper adapts existing methods to text data and explores practical considerations in their implementation.

Clarity: The paper is well-written, with clear statements of the requirements of the method and the assumptions it relies on.

Quality: The paper proves various propositions that guide the design of the proposed method in a principled manner. It also explores a falsification heuristic for untestable assumptions and empirically verifies that the heuristic is aligned with final performance in synthetic and semi-synthetic settings.

Significance: Unobserved confounding is a significant challenge in causal inference. The paper is well-motivated by the availability of large amounts of text data and the recent progress in language modeling.

**Weaknesses:**

- The significance of the empirical evaluation is limited due to synthetic and semi-synthetic experiments. The impact of the proposed method in realistic settings is hard to assess. While the evaluation of real settings is hampered by the lack of ground-truth, it may still be possible to synthesize data that is as realistic as possible so that the more realistic challenges of deploying such a method may be uncovered.

- The experiment design of the paper carefully avoids post-treatment text to avoid biased effect estimation. However, the proxy inference from language models indirectly depends on the pretraining dataset used to train the language model. The data in that pretraining set may be post-treatment text. The impact of this dependence is unclear.

- It is unclear how sensitive the performance of the proposed method is to the choice of pretrained zero-shot model, or whether we can expect better performance as the zero-shot model used scales up.

**Questions:**

As mentioned above:

- Do the authors expect the performance of their method to improve as they use larger language models to infer proxies?

- Can the authors provide any intuition for what they expect to be the impact of using a language model that has been trained on post-treatment text data to infer proxies and plug them into the causal effect estimation?

**Limitations:**

The paper sufficiently addresses limitations of the setting and method proposed.

---

> ### Author Rebuttal · Authors · 2024-08-06
>
> We thank the reviewer for the thoughtful feedback. We address the reviewer’s questions and comments on our paper's weaknesses.
>
> > The significance of the empirical evaluation is limited due to synthetic and semi-synthetic experiments. The impact of the proposed method in realistic settings is hard to assess. While the evaluation of real settings is hampered by the lack of ground-truth, it may still be possible to synthesize data that is as realistic as possible so that the more realistic challenges of deploying such a method may be uncovered.
>
> We thank the reviewer for this comment. We agree we should evaluate with the most realistic data possible. Yet, we believe our semi-synthetic experiments, which use real-world clinical notes from MIMIC-III, are the most realistic possible given our constraints.
>
> On Line 276, we write "In causal inference, empirical evaluation is difficult because it requires ground-truth labels for counterfactual outcomes of an individual under multiple versions of the treatment, data that is generally impossible to obtain Holland (1986); see Gentzel et al. (2019); Keith et al. (2023)." We plan to expand that writing to make the constraints of empirical evaluation in causal estimation even more clear. Gentzel et al. (2019) and Keith et al. (2023) describe methods for evaluation that are more realistic than semi-synthetic evaluation if one has an RCT dataset and can downsample the RCT dataset to create an observational (confounded) dataset. However, for the proximal setting we investigated in this paper, we could not find an existing RCT that would contain a sufficient amount of text and was suitable for the proximal causal inference set-up.
>
> In contrast, semi-synthetic experiments use real data for part of the DGP and then specify synthetic relationships for the remainder of the DGP. Semi-synthetic experiments have been used extensively for empirical evaluation of causal estimation methods; see [Shimoni et al., 2018](https://arxiv.org/abs/1802.05046); [Dorie et al., 2019](https://arxiv.org/abs/1707.02641); [Veitch et al., 2020](https://proceedings.mlr.press/v124/veitch20a/veitch20a.pdf). We will add additional writing clarifying our choice of semi-synthetic experiments in our camera ready paper upon acceptance.
>
> > Can the authors provide any intuition for what they expect to be the impact of using a language model that has been trained on post-treatment text data to infer proxies and plug them into the causal effect estimation?
>
> We thank the reviewer for this thoughtful comment. In Appendix B, we describe that if one of the proxies is constructed using post-treatment text and the other proxy is constructed from pre-treatment text, then proximal causal inference conditions could still be satisfied. However, we agree with the reviewer that if both proxies are constructed using LLMs that have post-treatment pretreatment text, this could be potentially problematic.
>
> **(1) We believe MIMIC-III is not contaminated in the pre-training datasets of the LLMs we use in our semi-synthetic experiments.**
>
> In our semi-synthetic experiments, we intentionally choose to use "open" LLMs, Flan-T5 and OLMo, in which the entire pretraining dataset is known and can be inspected. MIMIC-III is not available via a public URL (researchers have to sign-up and agree to a terms of service) so we do not believe it is contaminated in the pretraining data of either of these LLMs.
>
> Flan-T5 is built upon T5 which uses C4 (Colossal Clean Crawled Corpus) as its pretraining data (Raffel et al. 2020). Dodge et al. 2021 explore and document C4 and provide a search index, https://c4-search.apps.allenai.org/. We sampled full sentences from MIMIC-III and searched for them in C4 and did not have any search hits.
>
> OLMo is built from an open pretraining dataset Dolma (Soldaini et al. 2024), which is a 3 trillion token English corpus. Soldanini et al. describe that the data is acquired from sources that are "made accessible to the general public" such as the Common Crawl, GitHub, Reddit, Semantic Scholar, and Wikipedia. Again, because MIMIC-III is not publicly available, we again believe it is very unlikely any of the individual data is in the pretraining data.
>
> **(2) Other applications (e.g., social media) that have individual post-treatment texts in the pretraining data may need to rely on temporal time stamps.**
>
> Even though we have established that our semi-synthetic experiments using MIMIC-III may not be contaminated or contain individual data, there are other applications in which there could be post-treatment individual text data in the pretraining data of a LLM.
>
> For example, one might be using our proximal causal inference with text data design to infer proxies from social media posts from a platform such as Reddit. Individual posts that are post-treatment may possibly be scraped and included in the pretraining data of LLMs. In this scenario, to avoid this issue, we recommend (1) using open LLMs, such as we do with Flan-T5 and OLMo, in which the pertaining data can be inspected, and (2) using temporal time stamps to ensure that the LLMs pretraining data cutoff date is prior to treatment.
>
> > It is unclear how sensitive the performance of the proposed method is to the choice of pretrained zero-shot model, or whether we can expect better performance as the zero-shot model used scales up. [...] Do the authors expect the performance of their method to improve as they use larger language models to infer proxies?
>
> We thank the reviewer for this comment, and we will make this point even more explicit in the writing of our next draft. As the predictive performance of the zero-shot models improves, this will not affect the bias of causal estimates (the estimates will remain unbiased)  but it will **decrease variance (making confidence intervals narrower)**.

---

> > ### Comment · Reviewer_iLUt · 2024-08-10
> >
> > Thank you for the clarifications! I will maintain my score.

---

### Official Review · Reviewer_dab7 · 2024-07-13

**Soundness:** 2
**Presentation:** 2
**Contribution:** 2
**Rating:** 6
**Confidence:** 3

**Summary:**

The paper considers the causal effect estimation setting where a confounder is latent but with unstructured text data that could serve as proxies. Specifically, the paper proposes to incorporate zero-shot classifiers (to operate on text-based proxies), together with a falsification heuristic, into the proximal causal inference framework. The goal of the text-based proxy design is to craft W and Z that satisfy the identification condition in proximal causal inference framework. Empirical evaluations are presented in synthetic and semi-synthetic experiments.

---

**Post rebuttal**

I have increased my score, under the assumption that the revised manuscript could sufficiently address the original concerns on material organizations.

**Strengths:**

The strength of the paper comes from the attempt to conduct proximal causal inference on unstructured text data. The paper carefully considers identification conditions specified in the proximal causal inference framework, and proposes a design procedure together with a falsification heuristic to find two text-based proxies (so that the proximal inference can be conducted).

**Weaknesses:**

The weakness of the paper comes from the organization of the material (especially the constraints/conditions involved), and relatively simple settings considered in (fully-/semi-) synthetic experiments. In particular, further clarifications and/or discussions on following points would be very helpful (detailed in "Questions" section):

(1) regarding a series of different constraints, conditions, gotcha's, assumptions

(2) the fully and semi- synthetic experiments consider structured data, it is not exactly clear how these text proxies look like and if they correspond to all or part of the aforementioned constraints/conditions. Examples of text proxies would be much more intuitive than claiming "satisfy by design" (line 136)

**Questions:**

(1) regarding a series of different constraints, conditions, gotcha's, assumptions

In Section 2, there are P1 -- P4, a set of conditions should be satisfied in order for the proximal g-formula to work. In order to respond to the criticism of potential unavailability of W and Z in structured data, further assumptions S1 -- S2 are proposed. Then in Section 3 and Section 4, a set of gotcha's, two additional pre-conditions (lines 216 -- 218), and another set of conditions related to odds-ratio based heuristics (lines 245 -- 247) are presented. How do these conditions fit together to respond to the criticism of proximal causal inference on structured data? Are they all identification conditions for the text-based proxy proximal causal inference (other than the fact that they are conditions specified by previous works for different components put together)?

(2) the fully and semi- synthetic experiments consider structured data, it is not exactly clear how these text proxies look like and if they correspond to all or part of the aforementioned constraints/conditions

For fully and semi- synthetic experiments, the setting is largely structured data instead of text-based proxies. Examples of text proxies that the proposed design procedure yields would be much more intuitive than just claiming "satisfy by design" (line 136).

**Limitations:**

The organization of material could be improved to better present how the proposed approach responds to criticism of proximal causal inference (with structured data). Examples beyond fully and semi- synthetic experiments will make it clearer w.r.t. how the designed text-based proxies help to address the aforementioned criticism on previous work.

---

> ### Author Rebuttal · Authors · 2024-08-06
>
> Thank you for the reviewer’s comments on questions. We address each below.
>
> >[F]urther assumptions S1 -- S2 are proposed. Then in Section 3 and Section 4, a set of gotcha's, two additional pre-conditions (lines 216 -- 218), and another set of conditions related to odds-ratio based heuristics (lines 245 -- 247) are presented [...] How do these conditions fit together to respond to the criticism of proximal causal inference on structured data? Are they all identification conditions for the text-based proxy proximal causal inference (other than the fact that they are conditions specified by previous works for different components put together)?
>
> (P1) – (P4) are conditions for proximal causal inference as previously defined by Tchetgen Tchetgen (2020). The primary criticism of this work, as stated in lines 130-131, is that it is often difficult in practice to find structured proxies that fulfill (P1) – (P4).
>
> (S1) and (S2) are assumptions that limit the scope of our method. In lines 216-218, we further state two additional assumptions for our method: (1) the conditional independence of the two pieces of text and (2) W and Z being predictive of U. In Section 3, we show that (1) maps to (P1) – (P3), and (2) maps to (P4). These assumptions are not strictly necessary, and there may be other ways to satisfy the conditions for proximal causal inference, but we find these to be the most intuitive and easy for an analyst to verify/understand.
>
> In lines 245-247, we describe an odds ratio falsification heuristic: This is not an identification condition; rather,  it is an empirical check for violations of (P1) – (P4). As we show in our synthetic and semi-synthetic experiments, the falsification heuristic correctly flags potentially problematic proxies and prevents the practitioner from making incorrect downstream estimates.
>
> > For fully and semi- synthetic experiments, the setting is largely structured data instead of text-based proxies. Examples of text proxies that the proposed design procedure yields would be much more intuitive than just claiming "satisfy by design" (line 136).
>
> We kindly refer the reviewer to lines 292-313 as well as Appendices D, E, and F for descriptions of our semi-synthetic experiments. We use real-world clinical notes from the MIMIC-III dataset.
>
> We expand upon what we wrote in Section 5, and describe an example of how text proxies are obtained  with an example clinical note. Let us say that we are interested in obtaining text proxies for the latent variable atrial fibrillation. A clinical note may contain the following text:
>
>
> “The patient was suffering from breathing difficulties and irregular heart rate.”
>
> We then append a prompt to the clinical note and use it as input to a  zero-shot classifier (Flan-T5, for example):
>
> “Context: The patient was suffering from breathing difficulties and irregular heart rate.
>
> Is it likely the patient has atrial fibrillation?
>
> Constraint: Even if you are uncertain, you must pick either “Yes” or “No” without using any
> other words.”
>
> If Flan-T5 outputs “Yes”, the text proxy has a value of 1. If Flan-T5 outputs “No” or any other text, the text proxy has a value of 0.
>
> With more space in a camera-ready version, we are happy to include this example and the figure in the attached PDF file to our rebuttal in the paper to make this procedure more explicit.
>
> > The organization of material could be improved to better present how the proposed approach responds to criticism of proximal causal inference (with structured data). Examples beyond fully and semi- synthetic experiments will make it clearer w.r.t. how the designed text-based proxies help to address the aforementioned criticism on previous work.
>
> We thank the reviewer for this comment. Below is an example describing why proximal causal inference with structured data can be challenging. We had this example in  a previous draft but had to cut due to space constraints. With an extra page in the camera ready version, we plan to put this example back in to more clearly express our criticism of proximal causal inference with structured data:
>
> Suppose we aim to find proxies (in the structured variables) of atrial fibrillation (U) and have access to shortness of breath and heart palpitations. Although these proxies seem reasonable, we show how they violate the proximal causal inference conditions. Patient complaints about shortness of breath may affect which medication a clinician prescribes (A); hence, using shortness of breath as the proxy W violates (P2). Furthermore, a lack of oxygen resulting from shortness of breath may affect the healing of blood clots, thus influencing measurements of the D-dimer protein Y. In this case, using shortness of breath as the proxy Z violates (P3). Shortness of breath can also sometimes be a symptom of heart palpitations, violating (P1).

---

> ### Comment · Reviewer_dab7 · 2024-08-09
> **Thank authors for the response**
>
> Thank authors for the responses, and my concerns are largely resolved.
>
> While there is still some worry about the potential difficulty (upon readability, and upon the parsing of the key takeaway msgs) because of the conditions/assumptions/gothcha's introduced in various places, the proposed edits and material organization might help in the revised version of the paper. A brief paragraph (just like what authors presented in the response) or a table in the appendix would be helpful to provide a summary of how these requirements piece together.
>
> As for the examples, if the space permits, having some of them in the main paper might be very helpful for readers to bridge the gap between the set of requirements, and what can be done empirically (beyond fully- and semi- synthetic experiments).
>
> I have updated my evaluation score accordingly, under the assumption that the revised manuscript could sufficiently address the original concerns.

---

### Author Rebuttal · Authors · 2024-08-06

We thank the reviewers for their thoughtful comments and suggestions on improving our paper. We respond to each reviewer individually and address each reviewer’s questions point by point. We also include in our rebuttal a PDF with a figure that shows our proposed proximal causal inference with text design for our semi-synthetic pipeline (which uses real-world clinical notes from MIMIC-III). This figure specifically responds to reviewer dab7’s comment, and we intend to, given more space in a camera-ready version, add this figure to our paper.

If anything can be further clarified, please let us know. We look forward to further discussions.

---

### Decision · Program_Chairs · 2024-09-25

**Decision:**

Accept (poster)

**Comment:**

This paper treats causal inference in the setting where there is an "unobserved" confounder, but there is a proxy variable available in the form of text. The reviewers appreciated the problem setting, the principled approach, and the clarity of communication.